

# Benthic Silicon Cycling in the Arctic Barents Sea: a Reaction-Transport Model Study

James P. J. Ward[1], Katharine R. Hendry[1], Sandra Arndt[2], Johan C. Faust[3,4], Felipe S. Freitas[1], Sian F. Henley[5], Jeffrey W. Krause[6,7], Christian März[4], Allyson C. Tessin[8], and Ruth L. Airs[9]

[1]School of Earth Sciences, University of Bristol, Bristol, BS8 1QE, UK
[2]BGeosys, Department of Geosciences, Université libre de Bruxelles, Brussels, CP160/03 1050, Belgium
[3]MARUM - Center for Marine Environmental Sciences, University of Bremen, Bremen, 28359, Germany
[4]School of Earth and Environment, University of Leeds, Leeds, LS2 9JT, UK
[5]School of GeoSciences, The University of Edinburgh, Edinburgh, EH9 3FE, UK
[6]Dauphin Island Sea Lab, Dauphin Island, AL, USA
[7]School of Marine and Environmental Sciences, University of South Alabama, Mobile, AL, USA
[8]Department of Geology, Kent State University, Kent, OH, USA
[9]Plymouth Marine Laboratory, Prospect Place, Plymouth, PL1 3DH, UK

**Correspondence:** James Ward (JamesPJ.Ward@bristol.ac.uk)

**Abstract.** Over recent decades the highest rates of water column warming and sea ice loss across the Arctic Ocean have been observed in the Barents Sea. These physical changes have resulted in rapid ecosystem adjustments, manifesting as a northward migration of temperate phytoplankton species at the expense of silica-based diatoms. These changes will potentially alter the composition of phytodetritus deposited at the seafloor, which acts as a biogeochemical reactor, pivotal in the recycling of

key nutrients, such as silicon (Si). To appreciate the sensitivity of the Barents Sea benthic system to the observed changes in surface primary production, there is a need to better understand this benthic-pelagic coupling. Stable Si isotopic compositions of sediment pore waters and the solid phase from three stations in the Barents Sea reveal a coupling of the iron (Fe) and Si cycles, the contemporaneous dissolution of lithogenic silicate minerals (LSi) alongside biogenic silica (BSi) and the potential for the reprecipitation of dissolved silicic acid (DSi) as authigenic clay minerals (AuSi). However, as reaction rates cannot

be quantified from observational data alone, a mechanistic understanding of which factors control these processes is missing. Here, we employ reaction-transport modelling together with observational data to disentangle the reaction pathways controlling the cycling of Si within the seafloor. Processes such as the dissolution of BSi are active on multiple timescales, ranging from weeks to hundreds of years, which we are able to examine through steady state and transient model runs.

     Steady state simulations show that 60 to 98% of the sediment pore water DSi pool may be sourced from the dissolution

of LSi, while the isotopic composition is also strongly influenced by the desorption of Si from metal oxides, most likely Fe (oxyhydr)oxides (FeSi), as they reductively dissolve. Further, our model simulations indicate that between 2.9 and 37% of the DSi released into sediment pore waters is taken up with a fractionation factor of approximately -2 ‰, most likely representing reprecipitation as AuSi. These observations are significant, as the dissolution of LSi represents a source of new Si to the ocean DSi pool and precipitation of AuSi an additional sink, which could address imbalances in the current regional ocean Si

budget. Lastly, transient modelling suggests that at least one-third of the total annual benthic DSi flux could be sourced from



the dissolution of more reactive, diatom-derived BSi deposited after the surface water bloom at the marginal ice zone. This benthic-pelagic coupling will be subject to change with the continued northward migration of Atlantic phytoplankton species, northward retreat of the marginal ice zone and the observed decline in DSi inventory of the subpolar North Atlantic Ocean over the last three decades.

## 1 Introduction

The Barents Sea is one of seven shelf seas encircling the central Arctic Ocean and lies on the main inflow route for Atlantic Water. Oceanic circulation is driven by regional cyclonic atmospheric circulation and constrained by areas of prominent bathymetry (Fig. 1) (Smedsrud et al., 2013). Atlantic Water is fed in to the Barents Sea through the Barents Sea opening between mainland Norway and Bear Island. This water mass then flows northwards through the Barents Sea basin, where it is met by colder, fresher Arctic Water, infiltrating the Barents Sea from the northern openings (Oziel et al., 2016). The oceanic polar front delineates these two water masses, the geographic position of which is tightly constrained in the western Barents Sea by the bathymetry, but is less well defined in the east (Barton et al., 2018; Oziel et al., 2016). The heat content of the Atlantic Water-dominated region south of the polar front maintains a sea ice free state year-round, whereas the northern Arctic Water realm is seasonally sea ice covered, with a September minimum and a March/April maximum (Årthun et al., 2012; Faust et al., 2021).


The Barents Sea winter sea ice extent has been in decline since circa 1850 (Shapiro et al., 2003), but from 1998 the rate of retreat has become the most rapid observed on any Arctic shelf (Oziel et al., 2016; Årthun et al., 2012). Current forecasts suggest the Barents Sea will become the first year-round, sea ice free Arctic shelf by 2075 ($\pm$28 years) (Onarheim and Årthun, 2017). The atmospheric and water column warming driving this sea ice retreat is both a result of anthropogenic and natural processes, with recent 'Atlantification' arising from a northward expansion of the Atlantic Water realm (Årthun et al., 2012) and a reduction in sea ice import to the northern Barents Sea. The impact of these changes is an increase in upward heat fluxes, which inhibits sea ice formation (Lind et al., 2018).


The dynamic nature of the Barents Sea with respect to the physical oceanographic characteristics is reflected biologically in the ecosystems of the two main hydrographic realms. Annual primary production in the Barents Sea is estimated to range from 70 to 200 gC m$^{-2}$, with lower values found in the northern Arctic Water realm, where a deep meltwater-formed pycnocline limits nutrient replenishment through wind-induced mixing (Sakshaug, 1997; Wassmann et al., 1999). However, the most distinct peaks in the rates of primary production are found in the marginal ice zone (MIZ) (reaching 1.5-2.5 gC m$^{-2}$ d$^{-1}$, Hodal and Kristiansen (2008); Titov (1995)), which forms in spring/early summer as sea ice melts and retreats northwards, stratifying the water column and stabilising the nutrient-rich photic zone (Wassmann et al., 2006; Reigstad et al., 2002; Olli et al., 2002; Krause et al., 2018; Wassmann et al., 1999; Vernet et al., 1998; Wassmann and Reigstad, 2011). The phytoplankton communities of the Barents Sea in proximity to the polar front and MIZ tend to be dominated by pelagic and ice-associated diatom species, as well as the prymnesiophyte *Phaeocystis pouchetii* (Syvertsen (1991); Wassmann et al. (1999); Degerlund and Eilertsen (2010); Downes et al. (2021) and references therein).



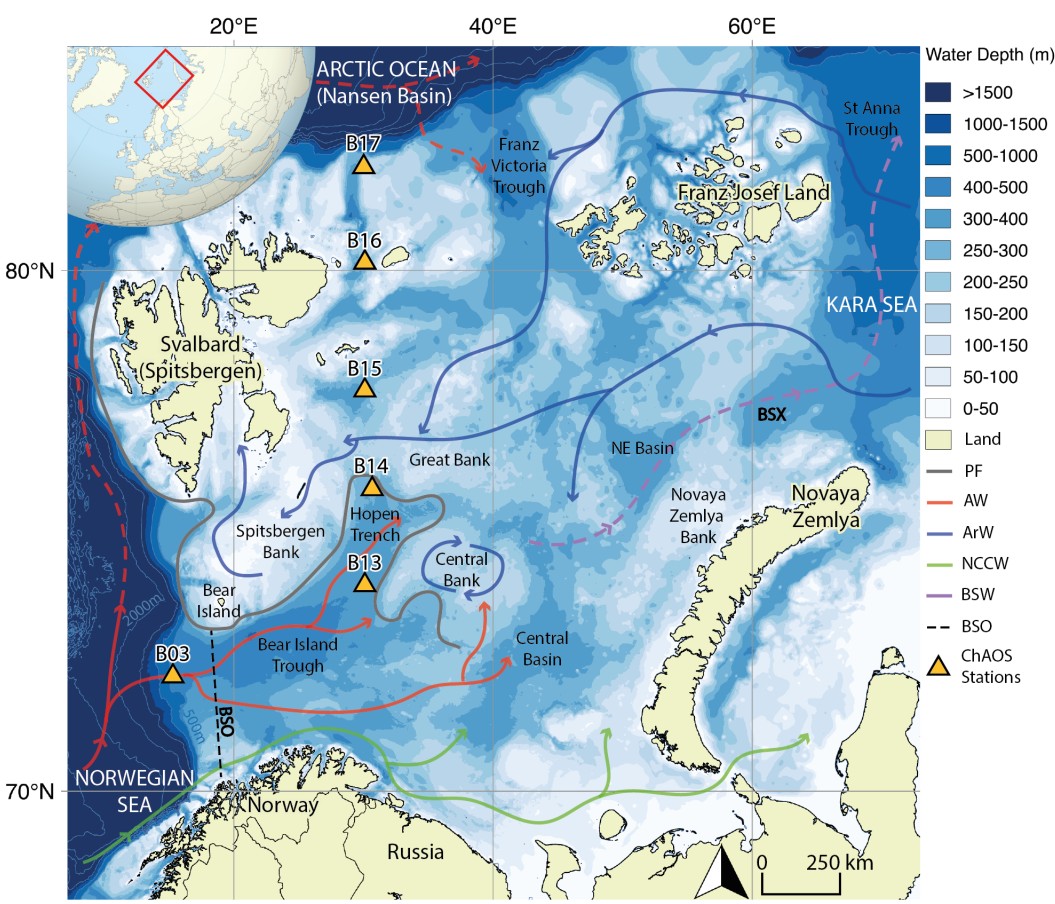

**Figure 1.** Map of the Barents Sea shelf, including ChAOS stations, main hydrographic features and main water masses (PF- oceanic Polar Front, AW- Atlantic Water, ArW- Arctic Water, NCCW- Norwegian Coastal Current Water, BSW- Barents Sea Water, BSO- Barents Sea Opening, BSX- Barents Sea Exit. Dashed red path refers to a subsurface water mass (Lien et al., 2013)). High resolution bathymetry data taken from the GEBCO global bathymetry model (Jakobsson et al., 2012).



Temperate flagellate species are becoming more dominant in the Eurasian Basin of the Arctic Ocean and are expected
to become the resident bloom formers in the region, with further expansion of the Atlantic Water realm and as sea surface
temperatures continue to rise (Neukermans et al., 2018; Orkney et al., 2020; Oziel et al., 2020; Ingvaldsen et al., 2021). Nutrient
concentrations in Atlantic Waters flowing into the Barents Sea have declined over the last three decades and are forecast to do
so throughout the 21st century (Neukermans et al. (2018) and references therein). Crucially, a much more significant drop in
dissolved silicic acid (DSi) concentrations has been observed relative to nitrate (Rey, 2012), creating less favourable conditions
for diatom growth (Neukermans et al., 2018). This shift in phytoplankton community composition is predicted to reduce the
export efficiency of organic material, with significant implications for pelagic-benthic coupling (Fadeev et al., 2021; Wiedmann
et al., 2020). Observations from long-term sediment trap data show that carbon export and aggregate sinking rates are two-fold
higher underneath diatom-rich blooms in seasonally sea ice covered areas of the Fram Strait, compared with that in *P. pouchetii*
dominated blooms in the ice free region (Fadeev et al., 2021). A similar contrast was observed in carbon export fluxes measured
using short-term sediment trap deployments north of Svalbard (Dybwad et al., 2021). It is estimated that 40-96% of surface
ocean primary production is exported to the seafloor in the Barents Sea (Cochrane et al. (2009) and references therein), while
the export efficiency of net primary production out of the euphotic zone in the central gyres is typically <10% (Turner (2015)
and references therein).

Diatoms take up DSi from seawater to build silica-based frustules (termed 'biogenic silica' (BSi) or 'opal'), which is then
recycled or reworked in transition to and within the seafloor. The seafloor acts as a biogeochemical reactor, generating a benthic
return flux of DSi across the pan-Arctic region that is estimated to equal the input from all Arctic rivers (März et al., 2015).
These recycling and reworking processes are therefore important for the regional Si budget and for fuelling subsequent blooms,
where seafloor-derived nutrients are able to be advected into the photic zone. We can then anticipate that the changes observed
in the predominant spring/MIZ bloom phytoplankton species and the efficiency with which phytodetritus is delivered to the
seafloor will impact the regional Si cycle, exacerbated by an observed decrease in the DSi concentration of the Atlantic Water
inflow over the last three decades (Rey, 2012; Hátún et al., 2017).

Given the changes forecast in the pelagic-benthic coupling of Si in the Arctic, it is important to understand the baseline
benthic biogeochemical system in order to anticipate the implications of further perturbations. Based on Si isotopic data from
various reactive sedimentary pools and the sediment pore water dissolved phase from the Barents Sea seafloor, Ward et al.
(under revision) hypothesised that the Si cycle is isotopically coupled to the redox cycling of metal oxides, most likely solid
phase Fe (oxyhydr)oxides. The reductive dissolution of Fe (oxyhydr)oxides and release of adsorbed Si (FeSi) is thought to
drive marked shifts in the isotopic composition of the Barents Sea sediment pore water DSi pool towards lighter compositions.
Further, Ward et al. (under revision) propose that sediment pore water undersaturation drives the contemporaneous dissolution
of lithogenic silicate minerals (LSi) alongside BSi, some of which is reprecipitated as authigenic clay minerals (AuSi), rep-
resenting a sink of isotopically light Si to the regional Si budget. Finally, Ward et al. (under revision) propose that seasonal
pelagic phytoplankton blooms generate stark peaks in pore water DSi that dissipate on the order of weeks-months. However, to
fully understand the early diagenetic cycling of Si within the seafloor of the Barents Sea we must be able to quantify the relative
contribution of LSi and BSi to the DSi pool, as well as establish whether AuSi precipitation removes a significant portion of





that pool. Here we employ steady state reaction-transport modelling to reconstruct the benthic cycling of Si in the Barents

Sea, informed by our dataset of solid and dissolved phase Si isotopic compositions (Ward et al. under revision) to test these hypotheses. Such techniques allow for the disentangling and quantification of the aforementioned early diagenetic reactions (Geilert et al., 2020; Ehlert et al., 2016a; Cassarino et al., 2020), as well as the return benthic flux of DSi to the overlying bottom water. Furthermore, reaction-transport modelling allows for the quantification of processes on much shorter timescales, thus we use transient model runs to validate the hypothesis that the pulsed deposition of bloom-derived BSi can perturb the

benthic Si cycle. We then quantify the bloom-derived BSi contribution to the total annual benthic DSi flux, the deposition of which is subject to the anticipated shifts in community compositions of pelagic primary producers across the Arctic Ocean.

Understanding the key aims presented here is not only important for anticipating the biogeochemical reponse of the Barents Sea seafloor to physical, chemical and biological changes in the surface ocean, but also has implications for the pan-Arctic Si budget. Currently there are disparities in the isotopic and mass balances of the Arctic Ocean Si budget, with Torres-Valdés et al.

(2013) concluding that the Arctic Ocean is a slight net exporter of Si. Furthermore, a recent isotopic assessment identified the need for an additional benthic sink of light Si to close the Si budget (Brzezinski et al., 2021). However, current understanding is limited by a lack of direct observations from major gateways, including the Barents Sea (Brzezinski et al., 2021). By coupling observational data with reaction-transport modelling we are able to construct a balanced Si-budget for the Barents Sea (section 3.5), contributing to the data gaps that currently limit our understanding of pan-Arctic Ocean Si cycling.

## 2   Methods

### 2.1   Reaction-transport modelling

#### 2.1.1   General approach

We use the Biogeochemical Reaction Network Simulator (BRNS) to disentangle the interplay of chemical and physical processes involved in the early diagenetic cycling of Si at stations B13, B14 and B15 of the Changing Arctic Ocean Seafloor

(ChAOS) project in the Barents Sea (Fig. 1 & 2, Table 1). These stations span the main hydrographic features (Polar Front) and realms (Atlantic and Arctic Water) of the Barents Sea. BRNS is an adaptive simulation environment suitable for large, mixed kinetic-equilibrium reaction networks (Regnier et al., 2002; Aguilera et al., 2005), which is based on a vertically-resolved mass conservation equation (equation 1) (Boudreau, 1997), simulating concentration changes for solid and dissolved species ($i$) in porous media at each depth interval and time step.

$$\frac{\delta \sigma C_i}{\delta t} = \frac{\delta}{\delta z}\left(D_{bio}\sigma\frac{\delta C_i}{\delta z} + D_i\sigma\frac{\delta C_i}{\delta z}\right) - \frac{\delta \sigma \omega C_i}{\delta z} + \alpha_i\sigma(C_i(0) - C_i) + \sum_j \lambda_i^j R_j, \qquad (1)$$

where $\omega$, $C_i$, $t$ and $z$ represent the sedimentation rate, concentration of species $i$, time and depth respectively. The porosity term $\sigma$ is given as $\sigma = (1 - \varphi)$ for solid species and $\sigma = \varphi$ for dissolved species, where $\varphi$ is sediment porosity (Table S2). This term ensures that the respective concentrations represent the amount or mass per unit volume of sediment pore water or solids as required (Boudreau, 1997). A full description of the model can be found in section S2 of the Supplementary Material.



**Figure 2.** Changing Arctic Ocean Seafloor (ChAOS) project summary. Chlorophyll-$\alpha$ represents the peak value measured at each station during JR16006 CTD casts, data available at doi:10.5285/89a3a6b8-7223-0b9c-e053-6c86abc0f15d. Benthic nutrient flux magnitudes are for DSi measured in this study and Ward et al. (under revision), as well as $PO_4^{3-}$ and $NH_4^+$ from Freitas et al. (2020). Red box is a schematic summary of the main processes involved in the early diagenetic cycling of Si in the Barents Sea. BSi reactivity values determined based on the steady state model simulations. Sea ice extent represents an approximation of the conditions at the time of sampling in 2017 and 2019. Please see supp. section S4 for a description of sediment pigment extraction methods. BSi- biogenic silica, LSi- lithogenic silica, AuSi-authigenic clay minerals, DSi- dissolved silicic acid.





### 2.1.2   Steady state reaction-transport modelling


Steady state modelling was employed to reproduce Si isotopic observational data, in order to quantify the reaction rates of key processes involved in the cycling of Si within the seafloor. The version of the Si BRNS model employed here is adapted from Cassarino et al. (2020), which largely follows the approach of Ehlert et al. (2016a) and assumes steady state. To ensure steady state was achieved in the baseline simulations, the applied run time was dependent upon the sedimentation rate (0.05-0.06 cm

yr$^{-1}$, Zaborska et al. (2008), Faust et al. (2020)) and core length, so as to allow for at least two full deposition cycles ($\sim$2500 years for a 50 cm Barents Sea core). The implemented reaction network accounts for a pool of pore water DSi, sourced by a dissolving BSi phase, from which Si can be incorporated into authigenic clay minerals (AuSi) as they precipitate. The kinetic rate law for the dissolution of BSi follows equation 2 (Hurd, 1972), where $k_{diss}$ is the reaction rate constant (yr$^{-1}$) and $BSi_{sol}$ is the solubility of BSi (mol cm$^{-3}$), implying that the rate of dissolution is proportional to the saturation state. The rate of

BSi dissolution is allowed to decrease exponentially downcore, in order to account for a reduction in reactivity due to BSi maturation and interaction with dissolved Al, as well as the preferential dissolution of more reactive material at shallower depths (Rickert, 2000; Van Cappellen and Qiu, 1997b; Rabouille et al., 1997; Dixit et al., 2001). The rate constants for BSi dissolution ($k_{diss}$, equation 2) were constrained using the solid phase BSi content measurements (Fig. 3).

Equation 2 represents a simplification of the reaction rate law, which in reality is influenced by processes not incorporated

into the model, such as surface area, temperature, pH, pressure and salinity. It is possible in some circumstances for the dissolution rate to deviate from the linear rate law (Van Cappellen et al., 2002), however, it is generally accepted that the dissolution of BSi is predominantly driven thermodynamically by the degree of undersaturation, leading to the linear rate law implemented in this study (Van Cappellen et al., 2002; Rimstidt and Barnes, 1980; Van Cappellen and Qiu, 1997b; Loucaides et al., 2012).

$$R_{db} = k_{diss} \cdot [BSi] \cdot \left( 1.0 - \frac{[DSi]}{BSi_{sol}} \right) \tag{2}$$

The precipitation of AuSi was modelled through equation 3, where $k_{precip}$ is the precipitation rate constant (Ehlert et al., 2016a). This rate law assumes that the reaction will proceed, providing the concentration of DSi is greater than the solubility of the AuSi ($AuSi_{sol}$). The rate is thus proportional to the degree of pore water DSi oversaturation (Ehlert et al., 2016a). We assume a value of 50 $\mu$M for $AuSi_{sol}$ at all three stations (Ward et al., under revision) (Table S2). As with BSi dissolution, the

rate of AuSi precipitation was allowed to decrease exponentially with depth, compatible with the hypothesis that the majority of AuSi precipitation occurs in the upper portion of marine sediment cores. Here, DSi can more easily precipitate in the presence of more readily available dissolved Al, the concentration of which is typically higher in the upper reaches of shelf sediments, sourced from the dissolution of reactive LSi (e.g. feldspar and gibbsite) contemporaneous to that of BSi (Aller, 2014; Rabouille et al., 1997; Van Beusekom et al., 1997; Ehlert et al., 2016a).

$$R_p = k_{precip} \cdot \left( \frac{[DSi]}{AuSi_{sol} - 1.0} \right) \qquad \text{if } [DSi] > AuSi_{sol} \tag{3}$$



In addition to the dissolution of BSi and precipitation of AuSi accounted for in previous early diagenetic modelling studies of the benthic Si cycle (Ehlert et al., 2016a; Cassarino et al., 2020), we incorporate the dissolution of LSi which is thought to be an important oceanic source of numerous elements, including Si (Geilert et al., 2020; Tréguer et al., 1995; Jeandel et al., 2011; Fabre et al., 2019; Ehlert et al., 2016b; Jeandel and Oelkers, 2015; Pickering et al., 2020; Morin et al., 2015). Here we

assume that the dissolution of LSi is predominantly driven by the degree of undersaturation (equation 4), although as with the dissolution of BSi, LSi dissolution is a complex reaction and sensitive to processes that are not included in the model, including the potential for being catalysed by microbes (Vandevivere et al., 1994; Vorhies and Gaines, 2009; Liu et al., 2017). The undersaturation of Si minerals is known to include most primary and secondary silicates, thus dissolution extends beyond BSi in marine sediments (Isson and Planavsky (2018) and references therein). Indeed, a suite of experiments have shown that

primary silicates and clay minerals can rapidly release Si when placed in DSi undersaturated seawater and take up Si in DSi enriched waters (Siever, 1968; Mackenzie et al., 1967; Lerman et al., 1975; Hurd et al., 1979; Fanning and Schink, 1969; Mackenzie and Garrels, 1965; Gruber et al., 2019; Pickering, 2020). Lerman et al. (1975) determined in one such experiment that the dissolution of eight clay minerals could be described by a first-order reaction rate law driven by the saturation state, consistent with that applied here. Further, our assumption that LSi dissolution is driven by the degree of undersaturation is

consistent with the suggestion that low bottom water DSi concentrations of the North Atlantic Ocean could allow for the dissolution of silicate minerals and thus account for high benthic DSi flux magnitudes in areas almost devoid of BSi ($<1$ wt%) (Tréguer et al., 1995). A value of $\sim100$ $\mu M$ was used for the solubility of LSi (LSi$_{sol}$) at all three stations, consistent with observations during multiple dissolution experiments of common silicate minerals in seawater (Table S3), as well as the estimated solubility of amorphous silica in high detrital component estuarine sediments (Kemp et al., 2021).

$$R_{dl} = k_{LSidiss} \cdot [LSi] \cdot \left(1.0 - \frac{[DSi]}{LSi_{sol}}\right) \qquad (4)$$

The desorption of Si from solid Fe (oxyhydr)oxide phases under anoxic conditions was simulated using a simple reaction rate constant ($k_{FeSi}$), representing the rate of desorption. The value assigned to $k_{FeSi}$ was calculated during the modelling exercise and no assumed amount of FeSi was included in the upper boundary conditions. This parameter likely represents a significant simplification, however the exact process pertaining to the adsorption of Si onto Fe (oxyhydr)oxides is unclear

and requires further study (Geilert et al., 2020). Step-functions were included in the FeSi reactions in the model to simulate the desorption of this phase at specific depth intervals, representing the Fe redox boundaries identified in Ward et al. (under revision). The step-functions act as a cut-off mechanism, either setting reaction rates to zero or activating them at specific depths. A full description of the model, including all boundary conditions and how isotopic fractionation was imposed in the AuSi precipitation and FeSi desorption reactions can be found in section S2 of the Supplementary Material.

Our estimates for all reaction rate constants in the steady state simulations ($k_{diss}$, $k_{precip}$, $k_{LSidiss}$ and $k_{FeSi}$) were not based on published values and were model-derived (Table S2). These values were constrained by ensuring best-fit of the observational data with the simulated solid phase BSi content and pore water DSi concentration and isotopic compositions, which were obtained by minimising the RMSE between simulated and measured values (Table S1). Despite being model-





derived, $k_{diss}$ values (0.0055-0.074 yr$^{-1}$) are found to lie within the published range for marine sediment BSi (section 3.2).
After the best-fit scenarios were established for each station, a sensitivity experiment was carried out by sequentially setting each reaction rate constant to zero, in order to assess the importance of each process to the model fit (Fig. 3).

### 2.1.3 Processing the simulated data

Depth-integrated rates ($R$) of a given reaction ($j$) were calculated across the model domain using equation 5, for the best-fit simulation data of each station.

$$R_j(x) = \sum_j \int_{x=0}^{x=L} R_j dx \tag{5}$$

where $L$ is the model domain length and $dx$ denotes the given depth interval.

   The deposition flux of BSi at the sediment-water interface (SWI) ($J_{BSi,in}$) was then calculated based on equation 6, which states that $J_{BSi,In}$ equates to the sum of the flux of BSi out of the sediment (assumed to equate to the integrated rate of BSi dissolution, $R_{db}$) and the BSi burial flux ($J_{BSi,bur}$) (Burdige, 2006; Freitas et al., 2021). $J_{BSi,bur}$ was estimated at the base of
the model domain (50.4 cm), following the mass accumulation equation 7 (Varkouhi and Wells (2020) and references therein), which is controlled solely by advection. The sedimentation rate at depth ($\omega_z$) was corrected for compaction following equation 8 (Berner, 1980). $J_{BSi,bur}$ calculations assume a sediment wet bulk density of 1.7 g cm$^{-3}$, consistent with previously assumed values for the Arctic seabed (Brzezinski et al., 2021; Backman et al., 2004) and that measured in clay-rich sediments of the Barents Sea (Orsi and Dunn, 1991).

$$J_{BSi,in} = R_{db} + J_{BSi,bur} \tag{6}$$

$$J_{BSi,bur} = (1 - \varphi_z) \cdot \omega_z \cdot [BSi]_z \cdot \rho_z \tag{7}$$

$$\omega_z = \omega_0 \cdot (1 - \varphi_0)/(1 - \varphi_z) \tag{8}$$

   We are also able to use the model simulation output to determine the total benthic flux of DSi at the SWI ($J_{tot}$), which has multiple constituent parts that contribute to the benthic flux magnitude. Following equation 9 (Freitas et al., 2020), we
calculate $J_{tot}$ and thus the relative contributions from bioturbation ($J_{bioturb}$), bioirrigation ($J_{bioirr}$), advection ($J_{adv}$) and molecular diffusion ($J_{diff}$), to complement the calculated $J_{diff}$ estimates and core incubation-derived $J_{tot}$ of Ward et al. (under revision).

$$J_{tot} = J_{bioturb} + J_{bioirr} + J_{adv} + J_{diff} \tag{9}$$





### 2.1.4 Transient reaction-transport modelling

The influence of seasonality in pelagic primary production on the benthic Si cycle of the Barents Sea has been inferred through interpretation of pore water DSi depth profiles at station B14 (Ward et al. under revision). However, BRNS assumes steady state and therefore cannot resolve seasonal biogeochemical dynamics without modification to allow certain boundary conditions to become time dependent, enabling their activation and deactivation on a temporal scale. Here we use transient model runs to test the hypothesis that the pulsed deposition of bloom-derived BSi can rapidly perturb the benthic DSi pool, which is then able

to recover on the order of weeks to months.

The steady state baseline simulations at station B14 represent a data-model best-fit of the 2018 observational data, wherein we do not observe transient peaks in sediment pore water DSi concentrations (Fig. 4). This steady state scenario was used as the initial conditions for the transient simulations, which were run for one simulation year, producing output data at weekly time intervals. We simulate the phytoplankton spring bloom event by incorporating an additional, more reactive BSi phase into the

model, coupled with a similar mechanism to the depth step-functions employed in the steady state reaction network, in order to initiate the deposition of the bloom-derived BSi pool in late spring and terminate it after one to three weeks. This time period is thought to represent the typical length of a Barents Sea MIZ and spring bloom respectively (Sakshaug, 1997; Dalpadado et al., 2020) (Fig. 4). All boundary conditions were kept constant, with the exception of those related to the bloom-derived BSi pool.

During the one to three week time interval the bloom-derived material was deposited at a rate equivalent to an increase in

the steady state BSi deposition flux of eight to 26-fold (Fig. 4). The background BSi deposition flux magnitude at station B14 is a model-derived parameter, constrained in the steady state simulations with the measured sediment BSi content and pore water DSi depth profiles. A 26-fold increase on the background deposition flux is consistent with observations of a 10 day post-bloom diatom mass sinking event in the sub-polar North Atlantic down to 750 m depth (Rynearson et al., 2013). The Barents Sea covers a relatively shallow continental shelf (average depth 230 m) and intense physical mixing at the polar front

has been shown to enhance rates of vertical organic carbon flux at depth, close to station B14 (Wassmann and Olli, 2004). We could therefore aniticpate an even greater increase in BSi depositional flux under bloom conditions than the maximum value assumed here. The reactivity of the bloom-derived material ($k_{dissbloom}$) ranged from 5 to 35 yr$^{-1}$, which is within the reactivity range of fresh pelagic BSi (3 to 100 yr$^{-1}$, Ragueneau et al. (2000); Nelson and Brzezinski (1997)) (Fig. 4). Each of these three boundary conditions (length of the bloom period, $k_{dissbloom}$ and the deposition flux) was varied across multiple

simulations within the constraints of published values to assess the influence of each parameter on the size and longevity of the sediment pore water DSi peak.

### 2.2 Observational data

As many reactions responsible for the biogeochemical cycling of Si between the solid and dissolved phases fractionate the isotopes of Si ($^{28}$Si, $^{29}$Si, $^{30}$Si) relative to each other, we are able to use stable Si isotopes as a tool to trace these pathways.

All solid phase, core top and sediment pore water samples collected for Si isotopic analysis were collected over three summers in the Barents Sea (30$^o$E transect spanning 74 to 81$^o$N) aboard the *RRS James Clark Ross* (2017, 2018, 2019). Dissolved





phase pore and core top water DSi concentration measurements were determined on-board using a Lachat QuikChem 8500 flow injection analyser with an accuracy of 2.8%, defined using CRMs (KANSO Co., Ltd.). Stable Si isotopic compositions of the samples were determined at the University of Bristol in the Bristol Isotope Group laboratory. Isotopic compositions are

expressed in $\delta^{30}$Si notation (per mille ‰), relative to the international Si standard NBS-28 (equation 10). A full description of field methods, as well as Si isotopic and concentration data of the solid and dissolved phase reconstructed using BRNS are provided in Ward et al. (under revision) (doi.org/10.5285/8933AF23-E051-4166-B63E-2155330A21D8).

$$\delta^{30}Si = \left( \frac{(^{30}Si/^{28}Si)_{sample}}{(^{30}Si/^{28}Si)_{NBS-28}} - 1 \right) \cdot 1000 \tag{10}$$

One early diagenetic process that fractionates isotopes of Si is the formation of AuSi clay minerals, which preferentially

takes up the lighter isotope from the dissolved phase, leaving sediment pore and core top waters relatively isotopically heavy in composition. The Si isotopic fractionation factor ($^{30}\epsilon$) associated with AuSi precipitation is relatively high (-1.8 to -2.2 ‰) (Hughes et al., 2013; Ziegler et al., 2005a, b; Opfergelt and Delmelle, 2012) and similar to that observed in the adsorption of Si onto Fe (oxyhydr)oxide minerals ($^{30}\epsilon$ -0.7 to -1.6 ‰) (Zheng et al., 2016; Delstanche et al., 2009; Wang et al., 2019; Opfergelt et al., 2009). However, the magnitude of $^{30}\epsilon$ associated with AuSi precipitation can increase with the number of

dissolution-reprecipitation cycles, which can reach -3 ‰ (Opfergelt and Delmelle, 2012). Similarly, $^{30}\epsilon$ during Si adsorption onto Fe (oxyhydr)oxides is thought to increase with mineral crystallinity ($^{30}\epsilon$ of -1.06 ‰ for ferrihydrite and -1.59 ‰ for goethite) (Delstanche et al., 2009). This characteristic is in contrast to the dissolution of BSi and LSi, which is assumed to occur without isotopic fractionation (Wetzel et al., 2014; Ehlert et al., 2016a), although previous work has shown that BSi dissolution could induce a slight fractionation that enriches the DSi pool in the lighter isotope ($^{30}\epsilon$ of -0.55 ‰) (Demarest

et al., 2009).

Model upper boundary conditions for the dissolved phase were based on the DSi concentration and Si isotopic composition of the core top waters (+1.64 ±0.19 (n=5), +1.46 ±0.15 (n=3) and +1.69 ±0.18 ‰ (n=6) at stations B13, B14 and B15 respectively). The Si isotopic compositions of the solid phases (BSi, LSi, FeSi) implemented in the model were determined based on a series of sequential digestion experiments carried out on surface sediment samples of the three aforementioned

stations (Ward et al. under revision). The digestion sequence (0.1 M HCl, 0.1 M Na$_2$CO$_3$, 4 M NaOH) activates operationally defined reactive pools of Si, including: Si-HCl (Si associated with metal oxide coatings on BSi, assumed here to represent the FeSi pool), Si-Alk (authigenically altered and unaltered BSi) and Si-NaOH (LSi and refractory BSi) (Pickering et al., 2020). The composition of the FeSi (-2.88 ±0.17 ‰, n=20) and LSi (-0.89 ±0.16 ‰, n=18) pools in Barents Sea sediment leachates were within long term reproducibility of Diatomite standard measurements (2$\sigma$ ±0.14 ‰, n=116) across the three stations,

whereas the BSi phase was found to vary spatially, with an isotopic composition of +0.82 ±0.16 ‰ (n=14) at station B15 and +1.43 ±0.14 ‰ (n=8) and +1.50 ±0.19 ‰ (n=7) at stations B13 and B14 respectively (Ward et al. under revision).



**Table 1.** Sampling station information averaged across the three Changing Arctic Ocean Seafloor (ChAOS) cruises.

| Station | Latitude ($^o$N) | Longitude ($^o$E) | Water Depth (m) | Bottom Water Temp ($^o$C) |
|---|---|---|---|---|
| B13 | 74.4331 | 29.9532 | 359 | 1.8 |
| B14 | 76.5019 | 30.5012 | 295 | 1.9 |
| B15 | 78.2192 | 29.9574 | 317 | -1.5 |

# 3 Results and discussion

## 3.1 What can reaction-transport modelling reveal about the controls on the background, steady state benthic Si cycle?

### 3.1.1 LSi dissolution and AuSi precipitation

An inverse modelling approach is applied here to reconstruct the benthic Si cycle of the Barents Sea to further investigate and disentangle the interplay of processes that combined to produce our observational dataset. Model results show that the sediment pore water DSi profiles cannot be reproduced by the dissolution of BSi alone at all three stations (B13, B14 and B15). At station B13 the simulations suggest that while there is sufficient DSi released to reproduce the asymptotic DSi concentration due to the higher BSi content at depth compared to B15, the rate of release in the upper sediment layers is not consistent with that in the pore water DSi concentration profiles downcore from the SWI in the observational data (Fig. 3). This observation is in contrast to B15, where the simulated asymptotic DSi concentration is just 23 $\mu$M when BSi is the only source of DSi, consistent with the measured BSi content profiles that suggest a cessation in BSi dissolution by the middle of the sediment cores ($\sim$15 cm, asymptotic BSi content of $\sim$0.2 wt%) (Fig. 3). Because of the continued increase in DSi with depth at station B13, partly driven by the elevated BSi content in the mid-core relative to B15, a relatively slow rate of AuSi precipitation is required at depth to take up the excess DSi and reproduce the observed asymptotic DSi value. Generally it is assumed that AuSi precipitation is concentrated in near-SWI sediments (0-5 cm), where the concentration of other essential solutes (Al, Fe, Mg$^{2+}$, K$^+$, Li$^+$, F$^-$) are generally highest, sourced from Fe and Al (oxyhydr)oxides and reactive LSi (Ehlert et al., 2016a; Van Cappellen and Qiu, 1997a; Mackin and Aller, 1984; Aller, 2014). However, the uptake of DSi through AuSi precipitation has previously been inferred in terrigenous-dominated shelf sediments of the Arctic Ocean at >50 cm depth (März et al., 2015).

Due to the discrepancies between observational and simulated DSi pore water concentration data, we incorporated a LSi phase into our model, which dissolves according to the presumed degree of undersaturation. Without this additional phase (when $k_{LSidiss}$ is set to zero), model simulations show that the rate of DSi release is insufficient to reconstruct the observational DSi data (Fig. 3). Implementing LSi dissolution in conjunction with BSi produced the best data-model fit. The dissolution of LSi has been inferred in a similar study of marine sediment pore waters of the Guaymas Basin (Geilert et al., 2020), as well as



**Figure 3.** Dashed red lines show best-fit steady state model simulations for stations B13 (top row) and B15 (bottom row), as well as additional fits based on a series of sensitivity experiments carried out to assess the importance of each reaction pathway. For station B14, see Fig. S2. From left to right: sediment pore water DSi concentration, sediment pore water DSi stable Si isotopic composition, solid phase BSi content. Vertical dashed black line represents the Si isotopic composition of the core top water from 2017 and open shapes show observational pore water data (Ward et al. under revision). Error bars on the Si isotopic data are based on long-term reproducibility, derived from repeat measurements of Diatomite Si standard ($2\sigma \pm 0.14$, n=116), unless that from analytical replicates was greater.



in beach and ocean margin sediments (Fabre et al., 2019; Ehlert et al., 2016b). Indeed, Morin et al. (2015) report Si dissolution rates for basaltic glass particles in seawater that exceed that of diatoms (Pickering et al. (2020) and references therein).

The inference based on the pore water DSi concentration profiles that an additional phase, most likely LSi, is dissolving into Barents Sea pore waters is supported by the simulated isotopic composition of the pore water phase at stations B13
and B15. Without the dissolution of the LSi phase, the $\delta^{30}$Si of the pore waters represents a mixture of the composition of the core top water and the BSi phase composition (+1.44 and +1.04 ‰ at B13 and B15 respectively) (Fig. 3). In this simulation scenario at station B15, the integrated rate of AuSi precipitation is zero as the concentration of pore water DSi has not surpassed the imposed AuSi solubility so cannot influence the sediment pore water $\delta^{30}$Si. Therefore, the model set up without LSi dissolution cannot reproduce the intricacies of the downcore isotopic profile, as the lack of DSi released results in
an insufficient concentration to allow for the precipitation of AuSi, thus the relative shift from isotopically lighter to heavier compositions between 0.5 and 2.5 cm cannot be resolved. The downcore shift to heavier isotopic compositions between 0.5 and 2.5 cm is thought to be caused by AuSi formation as the pore water DSi concentration crosses the saturation of the AuSi phase, facilitating its precipitation.

These model observations are consistent with a mass balance calculation using the isotopic compositions of the 0.5 cm pore
water sample, as well as the BSi and LSi leachate samples at stations B13 and B14, which indicate a contemporaneous release of both phases. Our model findings therefore support the hypothesis of Ward et al. (under revision) that LSi minerals are likely to be dissolving in the upper few cm of the Barents Sea seafloor. The depth-integrated reaction rates of the best-fit steady state simulations suggest that between 60 and 98% of the DSi released into the sediment pore water from the solid phase is sourced from the dissolution of LSi (Table 2). This range was determined by calculating the depth-integrated rate of LSi dissolution
across the model domain and three stations as a proportion of the total integrated rate of DSi input from the three simulated sources (BSi dissolution, LSi dissolution and desorption from metal oxides). The predominance of LSi over BSi dissolution is consistent with the observation that Barents Sea sediments consists of ∼96% terrigenous material (Ward et al. under revision), which is compatible with previous work showing that clay mineral assemblages in the Barents Sea are dominated by terrestrial signals from Svalbard and northern Scandinavia (Vogt and Knies, 2009). The Si isotopic composition of the LSi phase in
surface sediments of stations B13, B14 and B15 (-0.89 ±0.16 ‰, Ward et al. (under revision)) is also more consistent with that of dissolving secondary clay minerals (-2.95 to -0.16 ‰, Opfergelt and Delmelle (2012) and references therein) than primary silicates of the crust and mantle (∼0 and -0.34 ‰ respectively, Opfergelt and Delmelle (2012) and references therein). This isotopic composition is not surprising, given the predominance of the clay and silt size fraction in these sediment cores (87% <63 µm) (Faust et al., 2020).

However, while previous mass balance calculations and the model-derived integrated reaction rates presented here agree that both LSi and BSi dissolution contribute to the sediment pore water DSi pool, the magnitudes of the LSi dissolution contribution vary significantly. Ward et al. (under revision) suggest that just 14 and 13% of the DSi pool in the 0.5 cm pore water interval is sourced from the dissolution of LSi at stations B13 and B14 respectively, while at station B15 it is inferred that the $\delta^{30}$Si can be resolved through BSi dissolution alone. This is in contrast to the 84, 60 and 98% contributions calculated here (Table 2).
There are multiple contributing factors to this discrepency. Firstly, previous mass balance calculations are based on one depth





interval, whereas the estimates presented here are derived from depth-integrated dissolution rates of the entire 50.4 cm model domain. Furthermore, reaction-transport modelling has revealed that this contradiction is likely born of the assumption that the BSi pool at all three stations is sufficient to fuel the pore water DSi stock. Dissolution dynamics were not taken into account in the simple mass balance calculation of Ward et al. (under revision), however here we have shown that the composition of

Barents Sea surface sediments, which are almost devoid of BSi (0.26-0.52 wt%, or 92-185 $\mu$mol g dry wt$^{-1}$), cannot reproduce the rate of pore water DSi build-up with depth from the SWI and can only support an asymptotic DSi concentration of 23 $\mu$M at station B15 (Fig. 3). This observation implies that the additional assumption of Ward et al. (under revision) that the 0.5 cm pore water $\delta^{30}$Si value is not impacted by AuSi precipitation could be invalid. If the 0.5 cm pore water interval was directly or indirectly influenced by AuSi precipitation, such an assumption would lead to an underestimation of the LSi contribution, as

the $\delta^{30}$Si value would be isotopically heavier than if it were derived solely from dissolving solid phases mixing with trapped core top water.

Previously it has been suggested that the quantification of AuSi precipitation rates in marine sediments is not critical in order to fully understand the early diagenetic cycling of Si, as reverse weathering typically represents a diagenetic solid phase conversion from BSi to AuSi, via the dissolved phase (DeMaster, 2019). Model simulations reveal that 37, 2.9 and 13.8%

of the DSi released across the 50 cm model domain at stations B13, B14 and B15 respectively is taken out of solution in the formation of AuSi. This observation is consistent with a similar previous study of the Peruvian margin upwelling region, which determined that 24% of DSi released from the dissolution of BSi was reprecipitated as AuSi at a rate of 1.53 mmol Si m$^{-2}$ d$^{-1}$ (Ehlert et al., 2016a). AuSi precipitation rates in sediments of the Amazon Delta on the other hand can reach 7.7 mmol Si m$^{-2}$ d$^{-1}$ (Michalopoulos and Aller, 2004). Reprecipitation of the DSi pool within the relatively shallow cores studied here will

inhibit its exchange with overlying bottom waters. Therefore, in this context, AuSi precipitation can be considered a sink term for the regional Si budget (Ward et al. under revision). In a setting where AuSi precipitation occurs at such a depth where pore water DSi exchange with bottom waters is not possible, this reaction pathway could instead be considered an early diagenetic solid phase conversion (from BSi/LSi to AuSi), as opposed to a true sink term (Frings et al., 2016; DeMaster, 2019).

### 3.1.2 Evidence for coupling of the benthic Fe and Si cycles in the Barents Sea

Ward et al. (under revision) suggest that the benthic Fe and Si cycles are coupled in the Barents Sea, evidenced by a contemporaneous increase in pore water Fe concentrations with an enrichment in the lighter Si isotope of the DSi pool at all three stations. Model simulations support this hypothesis by demonstrating that the Barents Sea DSi pore water profiles can be reconstructed when applying the dissolution of both a BSi and LSi phase, however under the model scenario where the desorption of FeSi is inhibited ($k_{FeSi} = 0$), the $\delta^{30}$Si pore water profiles are inconsistent with the observational data (Fig. 3).

With the dissolution of the LSi phase implemented at both stations B13 and B15, it is possible to resolve the $\delta^{30}$Si pore water profiles in the upper 2.5 cm and 8.5 cm respectively. However, below these depths the simulated profiles have isotopically heavier compositions than the observational data. Release of an isotopically light phase at specific depth intervals (beginning at 1.5 cm at B13, 10 cm at B15) results in a simulated $\delta^{30}$Si profile within range of the observational data (Fig. 3). This isotopic shift to lower pore water $\delta^{30}$Si is interpreted to represent the desorption of Si from solid Fe (oxyhydr)oxide phases, wherein



**Table 2.** Depth-integrated reaction rates of the steady state best-fit simulations across the upper 50 cm of sediment (Fig. 3), as well as calculated BSi deposition rates and model-derived DSi benthic fluxes. All magnitudes are given in units of mmol Si m$^{-2}$ d$^{-1}$.

|  | Station | | |
| --- | --- | --- | --- |
| Parameter | B13 | B14 | B15 |
| AuSi precipitation ($R_p$) | 5.31 | 0.13 | 1.88 |
| LSi dissolution ($R_{dl}$) | 12.07 | 2.66 | 13.45 |
| Si desorption ($R_{FeSi}$) | 1.99 | 0.08 | 0.02 |
| BSi dissolution ($R_{db}$) | 0.29 | 1.69 | 0.15 |
| BSi deposition ($J_{BSi,in}$) | 0.32 | 1.71 | 0.17 |
| BSi burial ($J_{BSi,bur}$) | 0.027 | 0.025 | 0.021 |
| Total benthic flux ($J_{tot}$) | 0.14 | 0.34 | 0.25 |
| Diffusive benthic flux ($J_{diff}$) | 0.11 | 0.27 | 0.21 |
| % DSi sourced from LSi | 84 | 60 | 98 |
| % DSi reprecipitated | 37 | 2.9 | 13.8 |
| % BSi buried | 8.6 | 1.4 | 12.2 |

the depth intervals of the same isotopic shifts correspond to the depths at which Fe is released into the pore waters (Ward et al.; Faust et al., 2021). This increase in pore water Fe also occurs at similar depth intervals to decreases in pore water dissolved $O_2$ and $NO_3^-$ concentrations, consistent with a transition from oxic to anoxic conditions (Freitas et al. (2020); Ward et al. (in revision)).

As discussed above, the Si-HCl reactive Si pool is isotopically light and thought to be associated with metal oxide coatings
on BSi (Pickering et al., 2020). The $\delta^{30}$Si composition of the Si-HCl phase is assumed here to represent the composition of the phase desorbing across the Fe redox boundaries. When using a composition of -2.88 ‰ for the FeSi phase, simulation scenarios have identical DSi concentration profiles whether $k_{FeSi}$ is active or set to zero (Fig. 3). This similarity is in contrast to the $\delta^{30}$Si pore water profiles, which cannot be reproduced without release of this isotopically light phase at the redox boundaries of all three modelled stations.

This discrepancy between the sediment pore water DSi and $\delta^{30}$Si profiles when the FeSi desorption is active and inactive may explain why the influence of FeSi desorption is so apparent in Barents Sea sediment cores and more ambiguous in similar, previous studies of the Si cycle in lower latitude marine sediments. The preferential adsorption of $^{28}$Si onto Fe (oxyhydr)oxides and the subsequent dissolution or formation of these minerals has been used to interpret both heavy and light $\delta^{30}$Si marine sediment pore water signals in previous work (Ehlert et al., 2016a; Geilert et al., 2020). In addition, while not inducing a clear
signal in the $\delta^{30}$Si, redox cycling of Fe was highlighted as a potential regulating factor in the release of DSi into pore waters of the Greenland margin (Ng et al., 2020). Ng et al. (2020) hypothesised that the reductive dissolution of Fe mineral coatings increased the reactivity of the BSi pool, hence the elevated DSi concentrations found in cores with increased pore water Fe





(Ng et al., 2020). In the Barents Sea, FeSi desorption across sedimentary redox boundaries is thought to be so prominent in the $\delta^{30}$Si data because the asymptotic concentration of pore water DSi ($\sim$100 $\mu$M) is much lower than that in the aforementioned

studies ($\sim$350-900 $\mu$M). The low sediment pore water DSi concentration allows for the direct detection of this process, whereas in previous studies the influence of Fe on the benthic Si cycle is either inferred through elevated DSi and Fe concentrations (Ng et al., 2020), or by depositional context, for example in cores sampled from systems with an abundance of reactive Fe (e.g. hydrothermal vent systems (Fe sulfides) (Geilert et al., 2020) or the Peruvian oxygen minimum zone (Ehlert et al., 2016a)).

At stations B13 and B14, and to a lesser extent at B15, there is an increase in the sediment pore water DSi concentration

downcore from the middle to the base of the profiles. This feature can be reproduced in the model simulations with the desorption of FeSi from the respective redox boundary depths (Fig. S1), however the required ratio of $k_{FeSi}$ for $^{28}$Si and $^{30}$Si suggests an isotopic composition of the FeSi phase of just -1.0 to -1.5 ‰. This isotopic composition is heavier than that measured in the Si-HCl pool (-2.88 ‰), likely reflecting a complexity in the desorption process not captured by the model. Nevertheless, both scenarios support the release of an isotopically light phase at depth, most likely sourced from the Fe redox

cycle.

In summary, model simulations somewhat support the hypothesis, based on observational data, that the Barents Sea benthic Si cycle is influenced by the Fe-redox system. Model results suggest that the influence of the Fe redox cycle is relatively unimportant for the magnitude of the pore water DSi pool, which appears to be controlled by release from the BSi and LSi phases. However, the coupling of these element cycles is evidenced in the pore water $\delta^{30}$Si data, indicating that the Fe cycle is

important for the isotopic budget within the seafloor. We suggest the influence of FeSi desorption is detectable in the Barents Sea pore water $\delta^{30}$Si data, due to the relatively low pore water DSi concentrations and the distinctly isotopically light nature of the FeSi phase. These findings indicate that FeSi desorption should be considered when interpreting downcore $\delta^{30}$Si trends, especially in low DSi concentration settings.

### 3.2 Can seasonal phytoplankton blooms drive transient, non-steady state dynamics in the benthic Si cycle?

Observational pore water DSi concentration data from station B14 suggest that a pulsed increase in the deposition of reactive phytodetritus to the seafloor, derived from phytoplankton blooms, can drive transient peaks in pore water DSi of up to $\sim$300 $\mu$M (Ward et al. under revision). This non-steady state dynamic is evidenced in the three consecutive years of pore water DSi concentration data collected during the summers of 2017 to 2019, which show that in 2017 and 2019, when the MIZ was above station B14 just one and a half months prior to sediment coring, a sediment pore water DSi peak is present. This characteristic

is in contrast to 2018 when station B14 had been sea ice free for three months prior to sampling (Downes et al., 2021) and no peak in DSi concentration was observed in the pore water nutrient data (Fig. S6). This observation may indicate that the sediment pore water DSi peaks form under the MIZ, which supports the formation of phytoplankton blooms in late spring/early summer, supplying fresher BSi to the benthos relative to the background BSi pool. Dissolution of this fresher BSi could then fuel peaks in DSi concentration that dissipate between one and a half and three months after formation, driven by the enhanced

rate of molecular diffusion. Additional model simulations were carried out on the baseline, steady state best-fit model scenario of station B14 to assess this hypothesis.





Results of the transient simulations show that under certain conditions it is possible for the deposition of fresh, bloom-derived BSi to reproduce the observed peaks in pore water DSi concentration in 2017 and 2019. Calculated rates of background BSi deposition across the three stations (0.17 to 1.71 mmol Si m$^{-2}$ d$^{-1}$, Table 2) are similar to BSi export rates measured in short-

term and moored sediment traps in Kongsfjorden, Svalbard at 100 m depth and the eastern Fram Strait at 180-280 m depth (0.2-1.3 mmol Si m$^{-2}$ d$^{-1}$) (Lalande et al., 2013, 2016). A simulated three week, 10-fold increase of this BSi depositional flux at a much higher reactivity ($k_{dissbloom}$ 20 yr$^{-1}$) than the background value ($k_{diss}$ 0.074 yr$^{-1}$) derived from the 2018 baseline simulation, results in a DSi peak consistent with the magnitude of that in the observational data after one and a half months. This simulated peak in DSi concentration is then able to dissipate by three months after bloom initiation (Fig. 4). Similarly,

with a shorter bloom (one week), compatible with the typical length of an ice edge bloom in the Barents Sea (Dalpadado et al., 2020) and a 10-fold increase in BSi depositional flux of $k_{dissbloom}$ 15 yr$^{-1}$, the DSi peak is able to form and dissipate on a timeframe similar to that of the former scenario (Fig. 4). The generated peak in DSi concentration must be able to disperse after three months if the timing of core sampling relative to MIZ retreat is valid as the explanation for the lack of pore water DSi peak observed in the 2018 data.

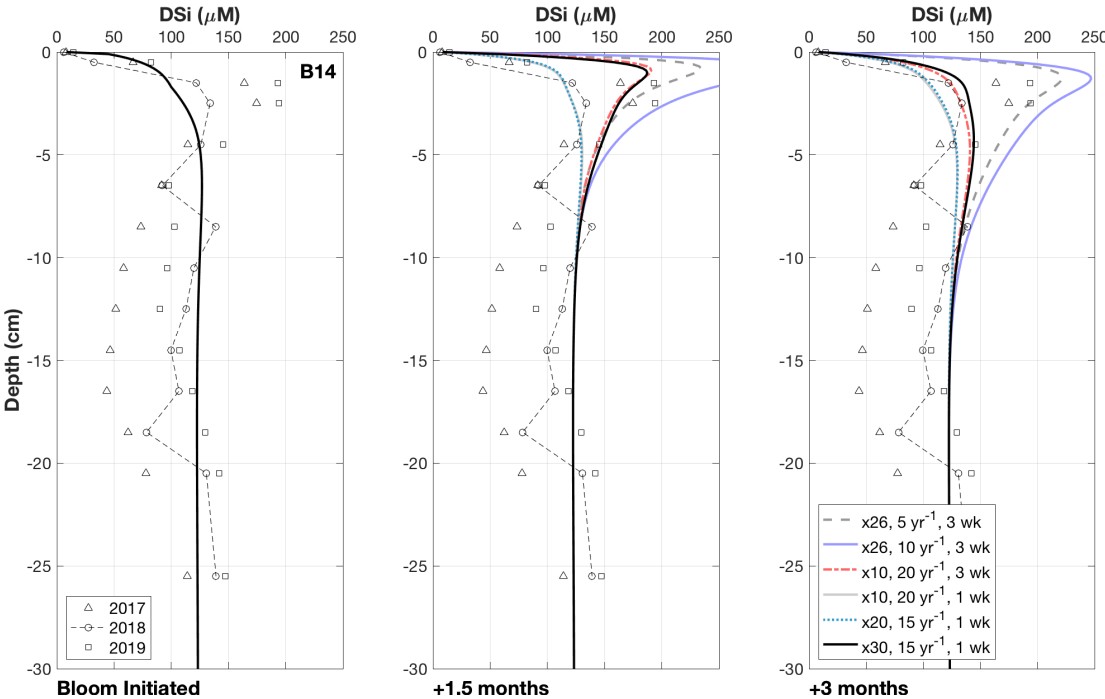

**Figure 4.** Results of the transient simulations under different conditions. Multiplier refers to the increase in the deposition flux ($J_{BSi,in}$) of the bloom-derived BSi, which was assigned a reactivity ($k_{dissbloom}$ in yr$^{-1}$) and deposited over either one or three weeks.





The implemented $k_{dissbloom}$ values fall within the reactivity expected of fresh diatoms, which is thought to range from 3 to 70 yr$^{-1}$ (Ragueneau et al., 2000). Dissolution rate constants of 0.98-1.38 yr$^{-1}$ have been observed in sediment cores collected from the Porcupine Abyssal Plain (mean water depth of 4850 m), considered high for BSi in deep marine sediments (Ragueneau et al., 2001), while values of up to 6.8 yr$^{-1}$ have been measured in much shallower sediments of Jiaozhou Bay in the Yellow Sea (Wu et al., 2015). These published values are lower than those employed in the transient simulations and

typically BSi material with dissolution rate constants of this magnitude are not observed in marine sediments. However, station B14 is located beneath the polar front, which is a location of intense physical mixing due to the interleaving of multiple water masses (Barton et al., 2018). This physical mixing was interpreted to be the driving force behind an enhanced particulate organic carbon depositional flux at ~150 m water depth, relative to that measured in sediment traps both north and south of the frontal zone (Wassmann and Olli, 2004). Furthermore, data collected from sediment traps deployed to the north and north-west

of Svalbard has uncovered an approximately two-fold higher vertical carbon export flux from diatomaceous aggregates formed in seasonally sea ice covered regions, compared with aggregates from *P. pouchetii* blooms (Fadeev et al., 2021; Dybwad et al., 2021). Therefore, it is considered here that given the shallow depth of the Barents Sea seafloor and the location of station B14 beneath the polar front, that fresh and reactive BSi formed in MIZ blooms could be efficiently ferried to the seafloor. It is likely that the production of BSi and its export from the surface ocean is highest on the shelves of the Arctic Ocean (Macdonald et al.,

2010). This observation, coupled with reduced BSi dissolution rates in the cold regional waters (Kamatani, 1982), results in more of the BSi recycling being transferred towards the seafloor (Brzezinski et al., 2021).

     The $k_{dissbloom}$ of the bloom-derived BSi is much higher than that required to reproduce the solid phase BSi content profiles of the background system ($k_{diss}$) (Fig. 3). At stations B13, B14 and B15, the implemented $k_{diss}$ values in the steady state simulations were 0.0055, 0.074 and 0.0105 yr$^{-1}$ respectively (Table S2, Fig. 2), which are not dissimilar to those estimated

for deep sea sediments from DSi pore water profile fitting procedures and flow-through reactor experiments (0.006-0.44 yr$^{-1}$) (McManus et al., 1995; Rabouille et al., 1997; Ragueneau et al., 2001; Rickert, 2000). The inverse of the dissolution rate constant provides an estimate of the residence time or mean lifetime of the given pool of BSi (McManus et al., 1995), which suggests that the less reactive pool of BSi has a mean lifetime of 182, 13.5 and 95 years at stations B13, B14 and B15 respectively. These mean lifetimes are too long to influence the Si cycle on a seasonal scale, which must be <1 yr$^{-1}$ to do so,

as is the case for organic matter (Burdige, 2006). The model-derived estimates of $k_{dissbloom}$ on the other hand would suggest a mean lifetime of approximately 20 days for the fresh BSi.

     Therefore, this work suggests that there are at least two types of BSi in Barents Sea sediments, one less reactive pool that dissolves at a slower rate and one fresher, bloom-derived pool that is able to perturb the sediment pore water DSi stock on a seasonal timescale. This conclusion is compatible with findings from a study of the equatorial Pacific region (McManus et al.,

1995) and observations from the Arabian Sea, indicating that the bulk sediment BSi content should not be treated as a single pool of uniform reactivity, but should instead be separated into reactive and unreactive fractions (Rickert, 2000; Schink et al., 1975). Consistent with these conclusions, Boutorh et al. (2016) observed up to an order of magnitude fall in $k_{diss}$ over the course of a three week batch dissolution experiment. This result indicates the presence of two phases of BSi within diatom frustules, one protecting the other, denoting a potential physiological basis for the differentiation in reactivity of seafloor BSi.





### 3.3 What is the simulated benthic DSi flux and how important is the contribution of bloom-derived BSi dissolution to the annual flux?

The simulated $J_{diff}$ magnitudes (0.11-0.27 mmol Si m$^{-2}$ d$^{-1}$) that contribute to $J_{tot}$ (Table 2) are within error of previously calculated $J_{diff}$ values (0.10-0.37 mmol Si m$^{-2}$ d$^{-1}$) for these Barents Sea sediment cores (Ward et al. under revision). Thus, our model-derived benthic DSi fluxes are well within range of a compilation of pan-Arctic shelf benthic DSi fluxes (-0.03 to +6.2 mmol Si m$^{-2}$ d$^{-1}$) (Bourgeois et al., 2017). Based on previous calculations and the simulation results presented here, we estimate that the mean DSi $J_{diff}$ benthic flux magnitude for the Barents Sea is +0.23 ($\pm$0.11 1$\sigma$) mmol Si m$^{-2}$ d$^{-1}$, ranging from +0.08 to +0.54 mmol Si m$^{-2}$ d$^{-1}$.

$J_{tot}$ at all stations is dominated by the molecular diffusive component ($J_{diff}$) (76-85%), in agreement with simulated estimates of phosphate fluxes at the same stations (Freitas et al., 2020) (Fig. 5). $J_{tot}$ at station B13 has the highest contribution from bioturbation (6%), consistent with the highest experimentally determined bioturbation diffusion coefficient of the Barents Sea stations (Solan et al., 2020). The advective component of $J_{tot}$ is negligible at all stations, while the bioirrigation element represents the greatest source of uncertainty in the simulated flux magnitudes, as this parameter was not constrained in parallel to this work, thus a global value was assumed (Thullner et al., 2009). At station B15, the model-derived $J_{tot}$ (+0.25 mmol Si m$^{-2}$ d$^{-1}$) is greater than $J_{BSi,in}$ (0.17 mmol Si m$^{-2}$ d$^{-1}$) (Table 2). This observation points to the release of an additional source of DSi to the dissolving BSi, which is compatible with the hypothesis that LSi is being released into the pore water dissolved phase.

DSi benthic flux magnitudes were also calculated for the transient simulations carried out on station B14 to quantify the influence of fresh bloom-derived BSi. Dissolution of the fresher BSi has an immediate and significant effect on the benthic flux, doubling the steady state background value of +0.34 mmol Si m$^{-2}$ d$^{-1}$ to +0.66 mmol Si m$^{-2}$ d$^{-1}$ within one week, which peaks two weeks after bloom material deposition at +1.84 mmol Si m$^{-2}$ d$^{-1}$, representing a five-fold increase (an additional +1.5 mmol Si m$^{-2}$ d$^{-1}$) on the background steady state benthic flux magnitude. The prominent DSi peak then dissipates and becomes largely undetectable after three months (Fig. 4). The average DSi benthic flux at the SWI over the twelve week period is +1.07 mmol Si m$^{-2}$ d$^{-1}$, indicating that the bloom-derived BSi releases an additional +0.73 mmol Si m$^{-2}$ d$^{-1}$ to the overlying bottom water. The steady state and transient model simulations therefore suggest that the background benthic flux of Si from the benthos is +124 mmol Si m$^{-2}$ yr$^{-1}$ at station B14, while the additional contribution over the twelve week period sourced from fresh BSi dissolution is +61.3 mmol Si m$^{-2}$ (based on a rate of +0.73 mmol Si m$^{-2}$ d$^{-1}$). This estimate suggests that a minimum of 33% of the total annual benthic flux of Si discharging from the seafloor at station B14 is sourced from the deposition of fresh BSi during the one week MIZ bloom.

The contribution of bloom-derived BSi dissolution to the annual benthic DSi flux magnitude reported here (an additional +0.73 mmol Si m$^{-2}$ d$^{-1}$ over the three months) for station B14 is greater than that in Ward et al. (under revision) (an additional +0.23 mmol Si m$^{-2}$ day$^{-1}$ over the same time interval), although the proportion is consistent across the two estimates (approximately one-third of the total annual benthic DSi flux). In part, this is due to the simulated fluxes incorporating the contribution from bioirrigation and bioturbation ($J_{tot}$). When using only the simulated $J_{diff}$ component, an additional +0.46 mmol Si m$^{-2}$





$d^{-1}$ is estimated to be sourced from the bloom-derived BSi, which is more consistent with the observational data calculations. However, this disparity is also due to the nature of the simulated flux calculation. The model-derived benthic flux magnitudes are calculated at the SWI, whereas previous $J_{diff}$ estimates are based on observational data of a much lower resolution, with the concentration gradient determined based on the DSi concentration in the core top water and in the sediment pore water at 0.5 cm depth. Furthermore, the simulated benthic flux estimates are based on a mean value derived from a weekly temporal resolution, which is not accessible in the observational data. However, both estimates can be used to draw a range of possible contributions from the bloom-derived BSi and although there is a disparity in the benthic flux magnitude, both methodologies suggest that at least one-third of the annual DSi benthic flux at station B14 is sourced from the dissolution of BSi deposited after a short MIZ bloom.

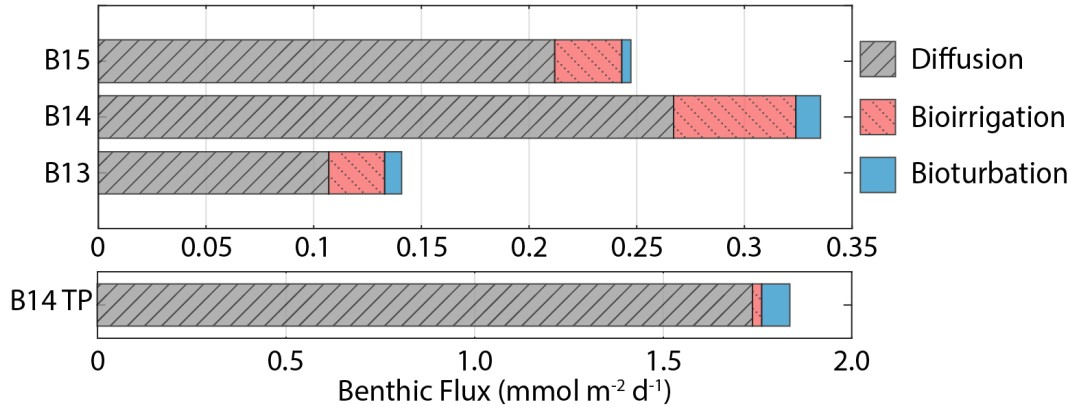

**Figure 5.** Benthic DSi flux magnitudes at the SWI calculated from steady state simulations. Includes contributions from molecular diffusion, bioirrigation and bioturbation. B14 TP refers to the B14 DSi flux magnitude at the peak of the transient simulations (x30 BSi depositional flux, 15 $yr^{-1}$ $k_{dissbloom}$, 1 week bloom duration).

### 3.4 How much BSi is buried long-term in the Barents Sea?

Traditionally, burial efficiencies of BSi were not included within Si budgets of the Arctic Ocean, due in part to a low mean BSi content ($<5$ wt%), as well as a low estimated sedimentation rate (a few mm $kyr^{-1}$) (Brzezinski et al., 2021; März et al., 2015). However, more recently Arctic Ocean seafloor BSi burial efficiencies are being re-evaluated due to revised models of sediment accumulation rates (Brzezinski et al., 2021). Asymptotic BSi content profiles approach 0.2 wt% at 14.5 cm below the SWI and remain constant with depth at all three Barents Sea stations (Fig. 3, Fig. S2), suggesting that some BSi is buried and able to accumulate in the seabed. Assuming a sedimentation rate of 0.05-0.06 cm $yr^{-1}$ (Zaborska et al., 2008), BSi particulates deposited at the SWI would approach 14.5 cm depth within approximately 240-300 years due to sediment accumulation, where BSi dissolution appears to cease. This timescale is a similar magnitude to the calculated mean lifetime based on $k_{diss}$ from the model simulations at station B13 (182 years), indicating that some BSi from this pool could accumulate at depth. However,





the steady state BSi pools at stations B14 and B15 have mean lifetimes of just 95 and 13.5 years respectively, theoretically precluding burial of this pool below 14.5 cm depth. The observation that BSi is present at $\sim$40 cm depth at all three stations with

a concentration of $\sim$0.2 wt% indicates that there may be three fractions of BSi constituting the bulk BSi phase: fresh bloom-derived BSi represented by $k_{dissbloom}$, the less reactive background BSi ($k_{diss}$), and a non-reactive pool that contributes to the 0.2 wt% BSi at depth alongside any surviving component from the background pool. A mean BSi burial rate of 0.024 mmol Si m$^{-2}$ d$^{-1}$ is estimated here for the Barents Sea (Table 2), corresponding to 0.012 Tmol Si yr$^{-1}$ for the whole shelf, assuming an area of 1.4 million km$^{-2}$.

The global ocean average BSi burial efficiency is $\sim$15% (Tréguer and De La Rocha, 2013), but values as low as 1-9% have been estimated in the outer Ross Sea, Pacific Ocean and north-west Atlantic Ocean (Ragueneau et al., 2001; DeMaster et al., 1996; Dale et al., 2021) and as high as 81-86% in the East China Sea (Wu and Liu, 2020) and Ross Sea (DeMaster et al., 1996). BSi burial efficiencies estimated here for three Barents Sea stations are low (1.4-12%, Table 2), but within range of the aforementioned published values. Our estimated burial efficiencies are based on the same sedimentation rates employed in the

model (Table S2), which are similar to the Barents Sea mean of 0.07 cm yr$^{-1}$ (Zaborska et al., 2008). However, Barents Sea sedimentation rates of up to 0.21 cm yr$^{-1}$ have been estimated since the last glacial period (Faust et al., 2021), which would significantly increase the calculated BSi burial efficiencies (5-33%) (equation 7). BSi burial rates at station B15 are the highest of the three stations and higher than the global average if a surface sedimentation rate of 0.21 cm yr$^{-1}$ is assumed (33%).

While the BSi burial efficiency calculated at station B14 is also within previously published values, it is much lower than the

Barents Sea stations to the north and south. Solid phase BSi contents in the surface intervals were determined from samples collected during the third cruise in summer 2019. As discussed, the pore water DSi profiles at B14 from this cruise are thought to be influenced by the dissolution of bloom-derived BSi, which would account for the elevated BSi content in the surface sediment relative to that at stations B13 and B15. The higher BSi content has likely resulted in an elevated estimation of $J_{BSi,in}$ and thus a reduced burial efficiency, which would also explain why the model-derived estimate of the contribution of

LSi dissolution to the DSi released from the solid phase is lower at station B14 (60%), due to an elevated $R_{db}$, enhanced by the deposition of fresher BSi. In addition, this would also explain why the estimated mean lifetime of the BSi from the steady state simulations at station B14 is much shorter than for the other two sites (13.5 years *vs* 95-182 years). If the measured surface BSi content at station B14 is indeed influenced by residual bloom-derived BSi, this would result in an overestimate of the background BSi reactivity in the model.

Relatively low burial efficiencies of BSi in sediments underneath oxygen depleted bottom waters (7-12%), similar in magnitude to those calculated here for the Barents Sea, have previously been attributed to low rates of bioturbation, resulting in less efficient export of BSi towards more saturated pore waters (Dale et al., 2021). Bioturbation coefficients were determined experimentally for the Barents Sea stations (2-6 cm$^{-2}$ yr$^{-1}$) (Table S2) and are much lower than might be expected, based on an empirical global relationship with water depth ($\sim$24 cm$^{-2}$ yr$^{-1}$) (Middelburg et al., 1997). Furthermore, the impacts of the

low rates of macrofaunal mixing on BSi burial efficiency is likely exacerbated by slow rates of sediment accumulation in the Barents Sea. High rates of BSi burial in the Bohai (60%) and Yellow Seas (42%) are thought to be driven by high sediment accumulation rates (Liu et al., 2002), which as with bioturbation is much lower in the Barents Sea than might be expected based



on an empirical global relationship with water depth (0.55 cm yr$^{-1}$) (Middelburg et al., 1997). The combination of low rates of macrofaunal mixing and sediment accumulation may therefore be the cause of the lower BSi burial efficiencies observed here
relative to the global ocean mean.

### 3.5  What are the implications of this work for the Arctic Ocean Si budget?

Brzezinski et al. (2021) uncovered an imbalance in the Arctic Ocean Si budget after carrying out an assessment using Si isotopes. The $\delta^{30}$Si values measured in the main Arctic Ocean water mass inflow and outflows are similar ($\sim$+1.70 ‰, Giesbrecht (2019); Liguori et al. (2020); Brzezinski et al. (2021)), suggesting that the cycling of Si within the Arctic Ocean has little
net effect on $\delta^{30}$Si. Brzezinski et al. (2021) conclude that given the relatively isotopically light input from fluvial sources (+1.30 $\pm$0.3 ‰, Sun et al. (2018)), as well as that emerging from seafloor sediments (+1.16 $\pm$0.11 ‰, Ward et al. (under revision)), balance must be maintained through the burial of isotopically light Si by BSi. However, a mass balance showed that the isotopically light inputs to the system are only partially offset by the burial of BSi (0.16-0.30 Tmol Si yr$^{-1}$), assumed to have a $\delta^{30}$Si of +1.16 $\pm$0.10 ‰ (Brzezinski et al., 2021). There must therefore be an additional sink of isotopically light Si
if the Arctic Ocean Si isotope budget is to maintain balance (Fig. 6). The absence of direct isotopic observations from some of the major gateways, including the Barents Sea shelf over which most of the DSi sourced from the Atlantic Ocean flows (Torres-Valdés et al., 2013), as well as a lack of data for the isotopic composition of BSi in Arctic Ocean sediments must be addressed to confirm the mechanisms proposed by Brzezinski et al. (2021). Si isotopes measured in the weak alkaline leachate (0.1 M Na$_2$CO$_3$, $\delta^{30}$Si$_{Alk}$) extracted from surface sediment sequential digestion experiments and measurements of $\delta^{30}$Si in
core top waters (Ward et al. under revision), coupled with reaction-transport modelling (this study) for stations B13, B14 and B15 contribute to the Arctic Ocean Si isotope dataset and help to fill these knowledge gaps.

$\delta^{30}$Si$_{Alk}$ in the Barents Sea ranges from +0.82 $\pm$0.16 ‰ at station B15 to +1.50 $\pm$0.19 ‰ at B14 (Ward et al. under revision). The Na$_2$CO$_3$ leachate activates an operationally defined reactive pool of Si, thought to be associated with authigenically altered and unaltered BSi (Pickering et al., 2020). Molar Al/Si ratios of the Na$_2$CO$_3$ support this concept, which fall within range of
that expected of BSi (Ward et al. under revision). $\delta^{30}$Si measured in the core top waters at the Atlantic (B13, +1.64 $\pm$0.19 ‰) and Arctic Water stations (B15, +1.69 $\pm$0.18 ‰) are similar to the composition of the main Arctic Ocean inflow and outflow water masses ($\sim$1.7 ‰) (Giesbrecht, 2019; Brzezinski et al., 2021; Liguori et al., 2020) and heavier than that measured in the BSi deposited at the seafloor. Therefore, assuming the composition of the BSi below the BSi dissolution zone within the seafloor is similar to that at the SWI, Barents Sea sediments represent a sink of $^{28}$Si, relative to the composition of the inflow
waters. However, $\delta^{30}$Si$_{Alk}$ at stations B13 and B14 is still isotopically heavier than the Arctic Ocean riverine input (+1.30 $\pm$0.3 ‰), as well as the assumed composition of the BSi buried across the Arctic seabed (+1.16 $\pm$0.10 ‰) in the Si budget (Brzezinski et al., 2021) (Fig. 6). Through reaction-transport modelling we have estimated that between 2 and 40% of the sediment pore water DSi pool is sourced from the dissolution of BSi. Moreover, 2.9-37% of the total amount of DSi released is reprecipitated as AuSi (Table 2). AuSi preferentially takes up the lighter isotope in the Barents Sea with a $^{30}\epsilon$ of -2.0 to -2.3
‰ (Table S2), thereby enhancing the preservation of BSi and further enriching the solid phase in the lighter isotope. Clays



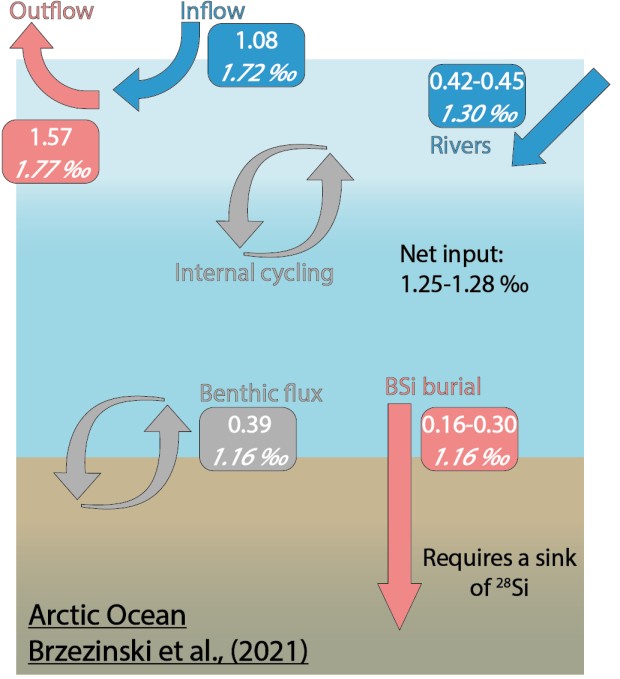
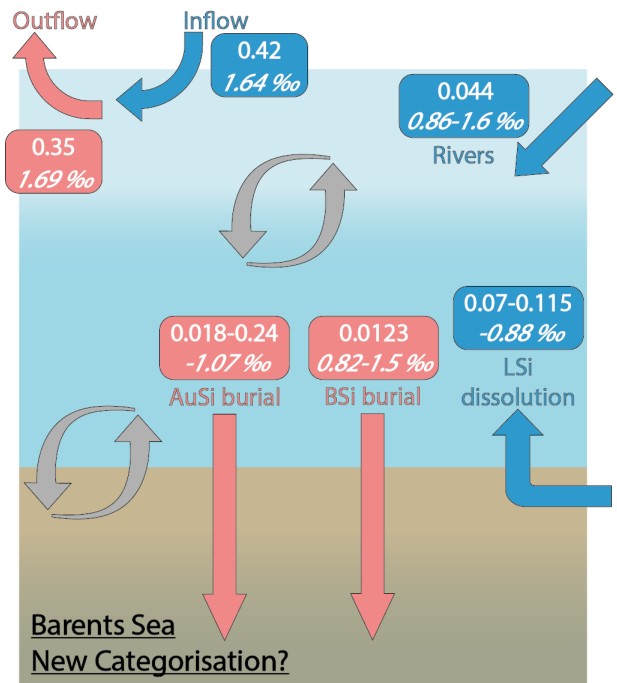

**In:**   0.11 to 0.16 Tmol Si yr⁻¹ ;  -0.39 to +0.08 ‰
**Out:**  0.03 to 0.25 Tmol Si yr⁻¹ ;  -0.97 to  -0.03 ‰

**Figure 6.** The Arctic Ocean Si budget (Brzezinski et al., 2021) (left) and a proposed Si budget for the Barents Sea (right), including a benthic flux recategorisation (i.e. contributions from BSi and LSi) and AuSi burial. Boxes include flux magnitudes given in Tmol Si yr$^{-1}$ (top values) and the flux $\delta^{30}$Si in ‰ (italicised bottom values). 'In' and 'Out' refer to the Si fluxes, discounting the water mass inflow and outflow (i.e. In (blue) = Rivers + LSi; Out (red) = AuSi + BSi). Grey boxes and arrows represent internal cycling. See supp. section 3 for further information on how the Barents Sea Si budget was calculated.

formed during weathering have a $\delta^{30}$Si composition ranging from -2.95 to -0.16 ‰ (Opfergelt and Delmelle, 2012). The burial of AuSi alongside BSi could therefore account for some of the isotopic imbalance.

Our reaction-transport model study has also highlighted the important contribution of LSi dissolution to the sediment pore water DSi pool (60-98%). If our findings on the dissolution of LSi are consistent across other Arctic shelves, a portion of the
benthic DSi flux cannot be defined as internal cycling of Si and should be recategorised as an additional input to that from the major ocean gateways and discharge from rivers. It is currently estimated that the benthic flux of DSi across the whole Arctic Ocean seafloor is ∼0.39 Tmol Si yr$^{-1}$ (März et al., 2015), therefore between 0.23 and 0.38 Tmol Si yr$^{-1}$ may represent an input of Si rather than a recycling term. This recategorisation could account for the additional Si inputs required to close the Si budget, as currently 32-47% (or 0.21-0.38 Tmol Si yr$^{-1}$) of the estimated net Si output is unaccounted for (Brzezinski et al.,
2021). The addition of AuSi as an output to resolve the isotopic imbalance would offset to some extent the release of DSi from LSi, while the LSi input compounds the isotopic imbalance identified by Brzezinski et al. (2021). However, here we show that





through AuSi precipitation acting as an additional sink for $^{28}$Si, both mass and isotopic balance can be attained in our proposed Si budget for the Barents Sea (Fig. 6, supp. section S3). Future work should look to assess whether similar relationships exist between the dissolution of LSi and precipitation of AuSi on other Arctic Ocean shelves, if we are to use these mechanisms to

balance the pan-Arctic Si budget.

## 4  Conclusions

In this study we quantify and disentangle the processes involved in the early diagenetic cycling of Si in the Arctic Barents Sea seafloor by reproducing Si isotopic and DSi concentration data from the solid and dissolved phase in a reaction-transport model. Baseline simulations are able to reproduce the observational data well, however we have also shown that the benthic Si

cycle is responsive on the order of days to the delivery of fresh BSi. Therefore, while the transient disturbances appear to be short-lived, future work should look to incorporate these processes into the baseline simulations.

Baseline model simulations also reveal that a significant proportion of the Si released from the solid phase within Barents Sea surface sediments is sourced from the dissolution of LSi (60-98%) on account of the low BSi contents (0.26-0.52 wt%). Furthermore, we demonstrate that without the influence of the Fe-redox cycle, which results in the release of Si adsorbed onto

solid Fe (oxyhydr)oxides under anoxic conditions, the observed isotopic composition of the pore water DSi pool cannot be reconciled. Both the LSi and FeSi sources are depleted in the heavier isotope (-0.89 and -2.88 ‰ respectively), as demonstrated in a sequential digestion experiment (Ward et al. under revision), consistent with the observation that sediments of the Barents Sea represent a source of light DSi to the overlying bottom waters (Brzezinski et al., 2021). Of the DSi sourced from BSi, LSi and FeSi, we show that between 2.9 and 37% is reprecipitated as AuSi. Coupled with the observation that a significant

proportion of the sediment pore water DSi pool is sourced from the dissolution of LSi, this hypothesis is significant for the regional Si budget. The dissolution of LSi represents a source of 'new' Si to the ocean DSi pool and the precipitation of AuSi inhibits exchange of pore water DSi with overlying bottom waters and therefore represents a sink term. These observations could require a recategorisation of a portion of the benthic flux in the Arctic Ocean Si budget, which is currently defined as a recycling term, as well as the inclusion of an additional Si sink. If LSi dissolution and AuSi precipitation is not exclusive to

the Barents Sea shelf, the additional input and isotopically light output could account for both the isotopic imbalance and the remaining proportion of net Si outflow that is currently unaccounted for (Brzezinski et al., 2021).

Model simulations also highlight a dichotomy in the cycling of Si in the Barents Sea seafloor, which is hypothesised to occur on at least two timescales. Observational data at stations B13 and B15 can be reproduced by assuming a steady state dynamic, thus representing a background system, which is controlled by the release of Si into the DSi pool from LSi and

the reprecipitation of DSi as AuSi. However, sampling across three years at station B14 has uncovered Si cycling on a much shorter timescale, controlled by the deposition of fresh phytodetritus. In this transient dynamic, the release of DSi is controlled by the dissolution of more reactive BSi. The processes occurring on the former steady state time frame will likely remain largely unaltered with further Atlantification of the Barents Sea, due to the mineralogical control on DSi release. Whereas the latter, transient system is reliant upon the seasonal delivery of fresh BSi, which is subject to change as the community



compositions of the MIZ and spring phytoplankton blooms shift to favour temperate Atlantic flagellate species (Neukermans et al., 2018; Orkney et al., 2020), or diatoms with lower silica content than polar species (Lomas et al., 2019). Furthermore, we have shown that the benthic DSi flux magnitude can increase five-fold after a simulated one week bloom, which is calculated here to contribute a minimum of one-third of the total annual flux of DSi from the seafloor at station B14. Any perturbation in the delivery of bloom-derived, relatively reactive BSi to the seafloor could therefore be detrimental to the total annual supply

of Si from Barents Sea sediments.

*Code and data availability.* Research data associated with this article can be accessed with https://doi.org/10.5285/8933AF23-E051-4166-B63E-2155330A21D8. Reaction-Transport Model code (Si BRNS) can be accessed with https://doi.org/10.5281/zenodo.6023767.

*Author contributions.* JF, FS, SH and AT helped in the sampling and processing of onboard samples. SH and JW measured Si concentrations, JW carried out Si isotopic analysis. JW and KH designed the sediment digestion experiments, carried out by JW. JF, CM and AT measured
pore water major and trace element concentrations and RA conducted surface sediment pigment analysis. SA, FS and JW developed the steady state and transient model code, JW performed the simulations. All authors contributed to data interpretation. JW prepared the manuscript with contributions from all authors.

*Competing interests.* The authors declare that they have no conflict of interest.

*Acknowledgements.* This work formed part of the Changing Arctic Ocean Seafloor project (ChAOS) of the Changing Arctic Ocean Pro-
gramme, funded by UKRI Natural Environment Research Council (NERC) (grant numbers NE/P005942/1, NE/P006108/1 and NE/P006493/1 2017-2022). We would like to express our gratitude to the captain and crew of *RRS James Clark Ross* for their support on cruises JR16006, JR17007 and JR18006. Additional logistical support was provided by National Marine Facilities and British Antarctic Survey. Further thanks go to colleagues for their support and advice, including C. Coath, L. Cassarino, J. Hatton, S. Bates, R. Ward, A. McAleer, R. Pickering, H.C. Ng, M. Puglini, J. Lucas, A. Stewart and J. Sharkey. Finally, we would like to thank the the editor and reviewers for their constructive
comments and feedback to help improve this article.



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
