# Peer review of "Benthic Silicon Cycling in the Arctic Barents Sea: a Reaction-Transport Model Study"

_Biogeosciences, 2022_

## Author Comment (AC1)

**Author Response to Comment on bg-2022-51**

Author Response to Anonymous Referee #1

Author Response to Referee comment on "Benthic Silicon Cycling in the Arctic Barents Sea: a Reaction- Transport Model Study" by James P. J. Ward et al., Biogeosciences Discuss., https://doi.org/10.5194/bg-2022-51-RC1, 2022

The manuscript by Ward et al. reports on the benthic-pelagic coupling in the Barents Sea with special emphasis on the Si cycle. The authors identified and described the biogeochemical reactions using silicic acid concentrations and Si isotopes and evaluated the reactions by reaction-transport modelling. The authors identified the dissolution of biogenic and lithogenic silica and silicon attached to dissolving iron phases as the major Si sources and authigenic clay precipitation as major sink. Also, the rapid dynamics and adjustment of the reactivity of the different sedimentary phases with respect to changing BSi supply and blooms are discussed and highlighted. Finally, the importance of benthic Si cycling for the Arctic Ocean Si budget is emphasized.

The manuscript is very well written and nicely discusses the main findings of this study. However, during reading the manuscript, I felt an increasing frustration with the many references to the other manuscript of Ward et al., which is currently under review in GCA. I appreciate that the authors provided a link to the preprint, but given that the method section (especially the sequential extraction and Si isotope measurements) and some parts of the interpretation and hypothesises are still under review leaves me with some concerns. In general, I am highly confident that the applied methods are correct and tested thoroughly, but I would only support a publication of this manuscript after the full review process and acceptance of the related GCA-manuscript. Apart from that, I am recommending this manuscript for publication with moderate revisions (see below).

***We thank the reviewer for their constructive comments and are happy to address the issues raised. The GCA manuscript has now been accepted for publication ([doi.org/10.1016/j.gca.2022.05.005](doi.org/10.1016/j.gca.2022.05.005)), which includes all the Si isotope measurements. The sequential sediment digestion protocol was published in Pickering et al., (2020) (doi.org/10.1029/2020GL087877).***

Introduction: I would start with a general introduction of the importance of the benthic silicon cycling as you have done in lines 69-78.

**Thank you for your suggestion. We have restructured the introduction to begin with the importance of the Si cycle and have moved some of the wider context into an oceanographic context section within the methods (lines 78-104).**

I guess, a reference to Fig. 2 is missing in the introduction! It is mentioned first time in line 449.

**Thank you for your comment. Fig. 2 is referred to on line 109 and we have added an additional reference on line 99 (please see the attached highlighted manuscript).**

Fig. 2: It is not clear at this point of the manuscript whether the reactions described in the red box are assumptions or data interpretation. Only later on in the text it becomes clear that these are modelling results.

**Thank you for highlighting this. We have specified in the Fig. 2 caption that the red box schematic is derived from the results of the simulations carried out in this study.**

Line 144: Instead of Ward et al., I would cite here the references you mention in the Table S2 (Lermann et al., 1975; Hurd, 1973).

**Thank you for your comment, the references on line 143 have been amended accordingly.**

Line 333-337: In this study, you discovered that some assumptions you made in your other study, which is also still under review, are not valid anymore. I would strongly recommend to use the possibility of changing the interpretation in your GCA manuscript, if you already know it is incorrect (concerning the AuSi precipitation in the upper 0.5cm)!

**Thank you for your comment. Without introducing the model into the observational data paper, it was not possible to introduce the alternative estimate for the contribution of LSi to the DSi pool. The model simulations build on and complement the observational data, but it was outside the scope of our GCA paper to incorporate the model results. Further, we have made multiple assumptions/simplifications in the model regarding LSi dissolution dynamics, as discussed in the text. Therefore, both the model and observational data-derived estimates provide two values for the contribution of LSi to the sediment pore water DSi pool, based on different methodologies, however both incorporate certain limitations/assumptions.**

Line 183: definition missing for RMSE

**Thank you for highlighting this, we have now included a definition for RMSE on line 182.**

Line: 250ff: for marine systems, no fractionation factor of authigenic clay formation is yet thoroughly established. The phrasing like it is sounds misleading. The studies you are referring to are either land-based, riverine or experimental. I agree that the size of the fractionation factor is likely correct, however, I would formulate this more carefully. Ehlert et al. (2016, GCA) modelled a fractionation factor of -2‰ for marine authigenic clay formation, which was also found in Geilert et al. (2020, Biogeosciences), but it can reach up to -3‰ in deep-sea settings (Geilert et al., 2020, Nat. Comm.), likely depending on pore water properties (pH, temperature, salinity, saturation states). This high fractionation factor would also agree with the repetitive number of dissolution-reprecipitation cycles discussed in Opfergelt & Delmelle (2012).

**Thank you for your suggestion. We have rephrased this section to reflect that the fractionation factors referred to are established for clay formation in riverine and terrestrial environments, rather than authigenic clay precipitation specifically (lines 276-293).**

Lines 275-340: it would significantly help, if you would refer to the model lines (colour, dashed, ...) shown in Fig. 3, when discussing the data. Like this, it is really difficult to connect the text with the various model results. Please also indicate in the legend in Fig. 3, what conditions cause the 'best fit'.

**Thank you for your suggestion, we have now referred to the plot line colours throughout sections 3.1.1 and 3.1.2 (dashed blue/red and solid grey/black) to improve readability. We have also specified in the caption of Fig. 3 that the best**

**fits require the dissolution of BSi and LSi, as well as AuSi precipitation and Si desorption from Fe (oxyhydr)oxides, and to refer to Table S2 for a full description of the boundary conditions imposed for the best fit simulations.**

Line 321: Considering the solubility of clays, can they really dissolve here? The dissolution rates of clays are much lower in seawater compared to primary minerals like feldspars or basaltic glass (see e.g. Jeandel & Oelkers, 2015).

**Thank you for highlighting this point. The authors acknowledge that the rate of dissolution for primary silicate LSi phases will be higher than for secondary phases. However, multiple experiments have demonstrated that clay minerals can rapidly release silica into DSi-depleted seawater (0.3-5 $\mu$M- not dissimilar to core top waters in this study (4-10 $\mu$M)) (some experiments were carried out at low temperatures of 1-2°C, most at room temperature. See Table S3), as well as take up DSi in DSi 'enriched' seawater (17-416 $\mu$M). Generally, these experiments show that a DSi concentration plateau is approached within four to 500 days (Fanning and Schink, 1969; Mackenzie et al., 1967; Lerman et al., 1975; Mackenzie and Garrels, 1965; Hurd et al., 1979), similar to asymptotic and pseudo-asymptotic concentrations in this study (~100 $\mu$M). Mackenzie and Garrels (1965) show a plateau after six months for six clay minerals, but note that >50% of the DSi contributing to the apparent solubilities was generally released within the first 10 days. Further, Köhler et al., (2003) (where the dissolution rates are taken for illite in Jeandel & Oelkers (2015)) show that at a low initial DSi concentration (5 $\mu$M) at 5°C, DSi concentrations reach 140 $\mu$M after just 140 days (although the solution was not seawater).**

**We also thank the reviewer for the link to Jeandel & Oelkers (2015), who suggest that clay minerals dissolve at a rate equivalent to ~0.1% of the clay (kaolinite and illite) mineral pool per year. Given the low rates of sedimentation estimated for the Barents Sea (average 0.07 cm/yr) (Zaborska et al., 2008), the length of the sediment cores (35 cm) represents ~500 years, corresponding to roughly half of all the clay material deposited at the sediment-water interface. Therefore, while the rate of dissolution is much lower than for primary minerals (e.g. apatite, labradorite, basaltic glass), over half of the material could theoretically dissolve across the sediment core lengths studied here. 96% of all the material at the sediment-water interface is estimated to constitute detrital material (Ward et al., GCA in press), therefore this would indicate over half of all the material could dissolve by the base of the core even at the lower rates of mineral dissolution.**

**In addition to the experimental settings discussed above, previous studies have inferred the dissolution of clay minerals in marine sediments (e.g. Geilert et al., 2020; Vorhies and Gaines, 2009; Abbott et al., 2019-doi.org/10.3389/fmars.2019.00504), and while clay minerals are typically considered a relatively stable end-product of weathering, fine clay particulates and reactive surface sites (e.g. montmorillonite, smectite, and illite) have been shown to dissolve in natural waters (Geilert et al., (2020) and references therein).**

**We have included a caveat on lines 350-351 to reflect that, while the $\delta^{30}$Si measured in the Si-NaOH pool (LSi) (-0.89‰, Ward et al., GCA (in press)) is within range of clay minerals (-2.95 to -0.16‰), rather than primary silicates (0 to -0.34‰), this value could also represent a combination of the two, thus potentially representing dissolution of reactive primary phases as well as clays (e.g. labradorite, basaltic glass or forsterite that dissolve at a rate of 20-40% per year (Jeandel & Oelkers, 2015)).**

Would it be possible that during your sequential leaching procedure you dissolved some of the authigenic clays here as well, shifting the bulk LSi phase $\delta^{30}$Si to lower values?

**The reactive Si pools studied here are operationally-defined (Pickering et al., 2020) and so it cannot be completely ruled out that some authigenic clay minerals have dissolved within the 4 M NaOH digestion, as there is inevitably some overlap across the pools. However, the 0.1 M HCl treatment is thought to remove authigenic products that coat BSi (e.g. Fe (oxyhydr)oxides and authigenic clays) (Pickering et al., 2020) and the mild alkaline (0.1 M Na$_2$CO$_3$) digestion activates the BSi phase as well as some LSi, which can then be corrected following Kamatani and Oku (2000) (Ward et al., GCA in press). Further, Michalopolous and Aller (2004) showed that mild acid leaches (0.1 M) dissolve authigenic clay coatings on BSi and note that poorly crystalline authigenic clays can also dissolve in distilled water within 24 hours (see references therein). The authors would therefore expect that the previous two digestions would have removed the highly reactive authigenic clays before the harsher alkaline digestion with 4 M NaOH.**

Lines 389-395: I wonder, if the model simulation gives a dissolving phase of -1 to -1.5‰, why not consider a higher contribution of lithogenic silica in this depth, which is much closer to the modelled value (about -0.9‰) than the FeSi phase (-about 2.9‰)? Do you really need a FeSi phase here to reproduce the pore water variability? I also wonder, if it is mass balance wise feasible? How much Si needs to be attached to this Fe-phase to create such a distinct peak in pore fluid $\delta^{30}$Si? And why is it then not seen in DSi?

**Thank you for your comment. In most sediment cores studied here, pore water DSi concentrations approach 80-100 $\mu$M at the depths where we see shifts towards lighter isotopic compositions at all three stations (Fig. 3). 80-100 $\mu$M equates to or surpasses the solubility of many primary and secondary silicate minerals in seawater at low temperatures (Table S3). Although, we recognise that some primary silicates in particular (e.g. olivine and some pyroxenes) can exhibit solubilities far greater than this (175-300 $\mu$M) (Hurd, 1979). The authors believe it is likely that the rate of LSi dissolution will slow considerably as Barents Sea sediment pore DSi concentrations approach equilibrium of many silicates (Lerman et al., 1975), especially given that Barents Sea sediments are clay-rich and so will likely lack considerable amounts of fresh primary silicates that would increase the average solubility of the bulk sediment.**

**Furthermore, in order for LSi dissolution to be the cause of the shift towards lighter Si isotopic compositions at specific depths at all three stations, a mechanism would be needed to explain a sudden increase in the rate of LSi dissolution within a particular sediment horizon. Given that the depths where we observe shifts towards lighter isotopic compositions also correspond to the shift to hypoxic/anoxic conditions (indicated by O$_2$ and NO$_3^-$ concentration data (Ward et al., GCA in press)) and corresponding increases in dissolved Fe concentrations, the authors believe the most likely driver is the reductive dissolution of Fe (oxyhydr)oxides.**

**If we assume an FeSi Si isotopic composition of -2.89‰, approximately 12-25% of the DSi in the sediment pore waters below the redox boundaries (where we observe the isotopic shifts) is composed of DSi derived from FeSi. 12-25% corresponds to 11-26 $\mu$M, which is not observed in the DSi concentration profiles, likely due to uptake during the precipitation of AuSi. Authigenic clay formation is thought to occur throughout the core lengths, as indicated by cation concentration profiles (Ward et al., GCA in press), although**

this process is likely enhanced near the sediment-water interface. This hypothesis is consistent with the simulations in this study, which require low levels of AuSi precipitation to continue at depth in order to generate best fits of the DSi concentration and isotope profiles (parameter 'ap' in Table S2). Providing the composition of the dissolving phase is isotopically light enough (e.g. -2.89‰) to offset the uptake of excess DSi through AuSi precipitation, this could explain why the process is seen in the isotope signals and not the DSi concentration profiles.

Section 3.2: Also here it would be easier to follow your arguments if you would refer to the colour coding of the model results in Fig. 4.

Thank you for your suggestion, we have now referred to the line colours throughout sections 3.1.1 and 3.1.2 (dashed blue/red and solid grey/black) to improve readability.

Figure 4: Why are the different scenarios in 'bloom initiated' only modelled for the x30, 15ye-1, 1wk scenario? Why not for the different multipliers, duration? Do you assume the bloom lasted only for one week as mentioned in line 426? In this case, I would add a comment in the caption as well.

Thank you for highlighting this. All the bloom scenarios were run in the 'bloom initiated' panel, however because this panel represents a very early timestep of the simulation, the model results are equivalent to the steady state simulations. The subsequent panels ('+1.5 months' and '+3 months') demonstrate the effect of the increased BSi deposition over time. We have now specified in the Fig. 4 caption that the 'bloom initiated' panel effectively represents time = 0 and so the different simulation results overlap. We have also specified in the caption that '+1.5 months' and '+3 months' refer to the time elapsed since the bloom was initiated for clarity.

Line 417: Which 'certain conditions' do you mean here?

Thank you for your comment. We have removed the phrase 'certain conditions' and referred to Fig. 4 instead (lines 444-445). This makes it clearer that the bloom-derived BSi can drive peaks in pore water DSi concentrations, depending on reactivity, rate of deposition and length of bloom (as described on line 234), which is not to be confused with the wider boundary conditions in Table S2.

Line 426: This combination of parameters does not exist in the legend in Fig. 4

Thank you for highlighting this error. We have changed the value to a 30-fold increase on line 453 (30X flux , 15 yr$^{-1}$ k$_{dissbloom}$ , 1 wk).

Line 525: The total ocean average BSi burial efficiency was revised in Tréguer et al., 2021 (Biogeosciences). The authors found a much higher burial efficiency compared to the findings of Tréguer & De La Rocha, 2013. How is that higher burial efficiency impacting your data interpretation?

Thank you for your suggestion. We have adapted lines 553-564, to reflect the updated burial efficiency from Tréguer et al., (2021). The burial flux magnitude is 46% higher in the 2021 review than the 2013 review, but the burial efficiency is similar: 3.6% vs 3% of the global marine gross BSi production. Here we report burial efficiency values as a proportion of the BSi deposited at the seafloor (11% calculated from Fig. 1 in Tréguer et al., (2021) and

calculated in Frings (2017) (doi.org/10.1007/s11631-017-0183-1)). Barents Sea stations therefore exhibit similar burial efficiencies to the global average, but are lower than most values reported for continental shelf sediments.

---

## Author Comment (AC2)

**Author Response to Comment on bg-2022-51**

Author Response to Anonymous Referee #2

Author Response to Referee comment on "Benthic Silicon Cycling in the Arctic Barents Sea: a Reaction- Transport Model Study" by James P. J. Ward et al., Biogeosciences Discuss., https://doi.org/10.5194/bg-2022-51-RC2, 2022

The manuscript by Ward et al. use previously published geochemical data (dissolved Si, d30Si; Ward et al. 2022) of sediment pore waters and solid phases from 3 stations in the Barents Sea in an attempt to better understand the reaction pathways controlling the cycling of Si at the seafloor, (e.g. the dissolution of LSi and bSi, coupling of Fe and Si cyles, and the potential precipitation of AuSi). They employ a reaction-transport model (steady state and transient) to try and understand the pathways controlling the biogechemical cycling of Si. The paper is mostly very well written, with some very interesting hypotheses (e.g. the possibility of being able to detect authigenic silica as a sink of silicon). However, the logic of the paper can be sometimes be difficult to follow, from the reader's perspective. There are some sections that are a bit confusing, and I have tried to provide some constructive feedback on the order of presenting the information (see below). My review includes a mixture of minor and major comments following the order of the text and sections of the manuscript.

*We thank the reviewer for their constructive comments and are happy to address the issues raised.*

**Line 16 to 17**– What do you mean by the phrase "taken-up" in this sentence? Are you suggesting that 2.9-37% of the released DSi has a value of -2 ‰?

*Thank you for your comment. We have reworded this sentence for clarity. In the best-fit model simulations, 3-37% of the DSi released from BSi and LSi dissolution is subsequently removed from the solid phase by a process that has a fractionation factor of ~-2‰, most likely representing authigenic clay mineral precipitation (lines 16-18).*

**Lines 144**– The assumed value of 50 mm for $AuSi_{sol}$ is not from Ward et al. it comes from Lermann et al. 1975, I believe.

*Thank you for highlighting this. Line 143 has been amended accordingly (please see the attached highlighted manuscript).*

**Figure 2.** It would be useful if the authors could indicate that the steady sate model simulations were from what is proposed in this paper.

*Thank you for your suggestion. We have amended the Fig. 2 caption to specify that the steady state simulations that contributed to the schematic are specifically from this study.*

**Section 2.1.4** – *Transient reaction-transport modelling*

I had a hard time following this section. Would it be possible to include a table or figure that could help the reader to understand the values associated with the input and outputs for this part of the model? For example, **lines 219 to 223 –** present an additional, more reactive BSi phase – but no data is provided. What does this look like in numbers?

*Thank you for your suggestion.* **We have rephrased the description on Lines 219-221 for clarity. The bloom-derived BSi is not an additional/new phase *per se*, instead this more reactive phase was modelled as a temporary increase in the deposition flux and reactivity of the BSi phase already incorporated in the reaction network. We hope that the further description of the transient boundary conditions (i.e. deposition flux, BSi reactivity and bloom timeframe) in the following paragraph is sufficient, alongside Fig. 4, in demonstrating what the bloom-derived BSi flux looks like in numbers across the bloom period and in describing how it is modelled.**

Also, it is mentioned **on line 225** that the conditions are either 1 or 3 weeks but that the deposition flux was -8 to 26-fold whereas in the figure 4, which is presented in the text before figure 3, The fluxes appear to be 10, 20, 30 and 26-fold.

*Thank you for highlighting this error. The text on line 226 has been amended (a 10 to 30-fold increase).*

Also, it is not clear why the figure presents 3 sub-figures (bloom, 1.5 months, 3 months) based on the detail provided in section 2.1.4. I had to read section 3.2 at the same times section 2.14. Please add more information to these sections and to the title for Figure 3.

*Thank you for your suggestion. In addition to the description in section 3.2 (lines 432-444), we have also included a brief description on lines 236-239 for readability of section 2.1.4. We have also expanded the Fig. 4 caption to further describe the three panels.*

**Section 2.2**

**Lines 249 to 260** appear to be disconnected from this section. It should be removed and/or perhaps placed in the discussion section.

*Thank you for your suggestion. This paragraph has been moved to section 3.1 as an introductory paragraph to the model results/discussion section (lines 276-293).*

Also, I wasn't aware that a fractionation factor of AuSi formation had been established nor that most people assumed that silicon isotope fractionation did not occur during dissolution. These are rather controversial points that should be presented more carefully.

**Thank you for your suggestion. We have rephrased this section to reflect that a fractionation factor for authigenic clay precipitation has not yet been established, and that the fractionation factors referred to in this paragraph are from studies of riverine and terrestrial environments (lines 276-289). We have also made our description of isotopic fractionation during BSi/LSi dissolution clearer (lines 289-294), by emphasising our point that some studies have shown that BSi dissolution occurs without fractionation, while others have demonstrated a slight enrichment of the lighter isotope in the dissolved phase. Consistent with previous reaction-transport model studies of the benthic Si cycle (Ehlert et al., 2016; Geilert et al., 2020), here we assume no fractionation during the dissolution of BSi/LSi.**

**Results and Discussion**

The authors provide a great deal of information in this section and it would be useful to have an introductory paragraph before section 3.1 to give a brief outline of what is to be expected as points of discussion. It is quite easy to get lost in the details provided. For example, an introductory paragraph could summarize the principle hypotheses that are to be discussed.

**Thank you for your suggestion. We have provided a short introduction for section 3 (lines 264-273), which summarises the goals/aims outlined initially on lines 48-75. We have also moved the section that discusses which early diagenetic reactions fractionate Si isotopes to section 3.1 (originally on lines 249-260).**

Along these lines, I am not quite sure why the authors did not choose to present the transient dynamics by phytoplankton blooms in section 3.1. I understand that this is not at steady-state, but it might be worthwhile to mention the possibility that the system is not at steady state. For example, on **lines 334-337,** the authors talk about dissolution dynamics and the lack of BSi in the Barents Sea. They could mention, in this section, that it is possible that the reaction-transport model is limited since it only operates under steady-state conditions, and then mention that they will discuss this further in section **3.2.**

**Thank you for your suggestion. We have added a brief discussion on lines 331-334 to highlight that the steady state model is limited as it cannot resolve all the dynamics at station B14 (e.g. the pore water DSi concentration profiles for 2017 and 2019). Evidence has not been found for transient dynamics at stations B13 and B15, and the isotopic composition and DSi concentration data can be reproduced by a model that assumes steady state. The authors therefore believe that B13 and B15 are at steady state, but could well be influenced by transient dynamics throughout the year that were not sampled due to a rapid recovery/return to steady state conditions. The authors have decided to separate the transient model discussion into its own section because of the lack of evidence for non-steady state dynamics at stations B13 and B15.**

Again, along these lines, it might be worthwhile to include in the title of section 3.2 that this still implies the use of the reaction-transport model...

**Thank you for your suggestion. We have amended the title of section 3.2 to refer to transient reaction-transport modelling.**

Regarding sections 3.1 and 3.2, I remain unconvinced by some of the arguments presented by the authors that the isotopically heavier signal (from 0.5 to 2.5 cm) is solely coming from LSi. I am not saying that it is not a plausible argument, but I wonder why the authors did not propose that benthic diatom activity could also be an explanation. Benthic diatoms have been found (alive) at incredible depths in the Barents Sea (Druzhkova et al. 2018), and they may be causing this observed shift in the top few cm of the sediment. At the very least it should be mentioned why this was not considered as a possibility. For example, the authors could make an argument after conducting a mass-balance calculation to show whether it is (or is not) possible.

**Thank you for your suggestion. Druzhkova et al. 2018 observed three species of autochthonous benthic diatoms at up to 245 m water depth in the central Barents Sea, with cell counts of 67-365 cells/cm². Previously reported DSi quotas per diatom cell typically range from 0.01-1 pmol/cell (Li et al., 2016-doi.org/10.5194/bg-13-6247-2016; Olsen and Paasche, 1986-doi.org/10.1080/00071618600650211; Lomas et al., 2019-doi.org/10.3389/fmars.2019.00286). For the following mass balance calculation, we assume the maximum cell quota (1 pmol/cell) and cell count (Druzhkova et al. 2018), as well as an 8 $\mu$M average core top water DSi concentration for a theoretical area of 1 m² and 2 cm deep (0.02 m³) at the sediment-water/core top interface.**

The aforementioned values give a DSi stock in the theoretical volume of 160 $\mu$mol DSi and a maximum of 3.7 $\mu$mol DSi stored in the diatoms across that theoretical space. Modelling this system (at station B14 for example) under a steady state scenario (see equations in Grasse et al., (2021)- doi.org/10.3389/fmars.2021.697400), assuming an average fractionation factor during DSi uptake by diatoms of -1.1‰ (De La Roach et al., 1997; Sutton et al., 2013) and an initial $\delta^{30}$Si of 1.17‰ (equivalent to the average 0.5 cm pore water composition across the three cruise years), gives a predicted $\delta^{30}$Si of 1.19‰, which is much lower than the measured average isotopic composition at the 1.5-2.5 cm peak (1.48‰) (maximum f = (160 $\mu$M - 3.5 $\mu$M)/160 $\mu$M = 0.98). A Rayleigh fractionation scenario also gives a predicted $\delta^{30}$Si at the pore water peak of 1.19‰, under the same boundary conditions.

These calculations indicate that the autochthonous benthic diatoms are not present in a high enough abundance to be solely responsible for the shift towards heavier isotopic compositions downcore from 0.5 cm to ~2.5 cm sediment depth. This observation is consistent with our previous findings, which show that Barents Sea surface sediments are characterised by low BSi contents (0.26-0.52 wt%). Furthermore, the depth of the stations studied here range from 295-359 m, which is greater than the depth at which the autochthonous benthic diatoms were found, suggesting they may exhibit an even lower abundance at stations B13, B14 and B15.

We have added lines 327-330 to the main text to acknowledge the presence of benthic diatoms in the Barents Sea, that are likely to be taking up DSi, but that they are not in a high enough abundance to influence the isotopic composition of the sediment pore water DSi pool to a significant degree. These observations, in addition to the evidence presented in Ward et al., GCA (in press), support the precipitation of authigenic clay (AuSi) as the driver of the downcore increase in pore water $\delta^{30}$Si between 0.5 cm and 2.5 cm depth across the three stations.

Ideally, it seems as though it would be helpful to conduct empirical assessments (batch or open conditions) of the dissolution of sediments to test whether the hypotheses are plausible for the dissolution of LSi, BSi, and the links between the Fe and Si cycles over time. Since very little work has been done really evaluating these aspects, it would be very useful for the authors to suggest the need for more empirical studies in order to support (or not) the model results from this work.

Thank you for your suggestion. We have expanded our recommendations for future work at the end of section 3.5 by including batch and/or flow-through experiments to investigate dissolution (BSi/LSi) and precipitation (AuSi) dynamics, as well as the coupling of the Fe and Si cycles (lines 631-633).

Figure 3. I do not see the interest in showing a model of the BSi content, in particular since it is based on only 3 data points per station.

Thank you for your comment. The BSi content was a key parameter for the fitting of our observational data, used to constrain the boundary conditions for the dissolution of BSi as opposed to assuming published values for the reactivity or exponential attenuation function. The authors have therefore decided not to remove the BSi content panel in Fig. 3.

Lines-434 to 435: why does your simulation data ($k_{dissbloom}$) have higher values than the published data, even higher than the dissolution rate constant of diatom at warm temperature (14-22 degrees)? The reactivity of fresh diatoms varies due to

temperature: high reactivity of diatoms has been observed at a higher water temperature region, whereas low reactivity of diatom material was observed in cold water, and the differences can be more than 10-fold (Ragueneau et al., 2000). Therefore, the reactivity of diatom bSi in your modeling of the Arctic area (<2 degrees) might be much lower than what was used in the model.

**Thank you for highlighting this. We have amended lines 460-466 to acknowledge the sensitivity of BSi reactivity/dissolution rate constants to in-situ temperature and species composition. The reactivity of fresh diatoms is thought to range from ~3-100 $yr^{-1}$ (Ragueneau et al. 2000; Nelson and Brzezinski, 1997), although Roubeix et al., (2008) suggest that at 18-23 °C, dissolution rate constants range from 1.5 to 150 $yr^{-1}$ across different diatom species. Our modelled scenarios cover a range of 5-20 $yr^{-1}$, which is also within range of the reactivity of fresh/cleaned diatoms measured at much lower temperatures in seawater (27 $yr^{-1}$ at 2°C (Rickert, 2000) and 6 to 131 $yr^{-1}$ at 4°C in Kamatani and Riley, (1979) depending on the stage of dissolution).**

**The authors recognise that BSi reactivity can decrease with a reduction in temperature (e.g. 4 to 14-fold lower at -2 *vs* 12 °C (Tréguer et al., 1989-doi.org/10.1007/BF00442531)), as well as across different species. The water column BSi dissolution temperature coefficient ($Q_{10}$) is ~2.3 - 2.9 (Kamatani, 1982- doi.org/10.1007/BF00393146 ; Bidle et al., 2002-doi.org/10.1126/science.1076076 ; Natori et al. 2006-doi.org/10.1016/j.marchem.2006.04.007), thus the BSi dissolution rate at 0 °C would theoretically be 5 to 8.4-fold lower than the rate at 20 °C. However, previous experiments have shown that dissolution rate constants much greater than those needed to resolve the transient/non-steady state dynamics at station B14 have been observed at low temperatures. We suggest that the physical mixing dynamics at the polar front (station B14) are able to efficiently deliver fresh, reactive BSi to the relatively shallow Barents Sea seafloor (as it has been shown to do for organic carbon (Wassmann and Olli, 2004)), which rapidly dissolves and drives stark transient peaks in pore water DSi concentrations of the uppermost sediment layers.**

**Line-457:** study carried out by Moriceau et al., 2009 reported at least 2 types of bSi, this reference may also be relevant to your study.

**Thank you for your suggestion. We have included this reference and amended the text on lines 489-492.**

**Line 525:** please use the updated values for burial efficiency provided by Tréguer et al. (2021)

**Thank you for your suggestion. We have adapted lines 553-564, to reflect the updated burial efficiency from Tréguer et al., (2021). The burial flux magnitude is 46% higher in the 2021 review than the 2013 review, but the burial efficiency is similar: 3.6% vs 3% of the global marine gross BSi production. Here we report burial efficiency values as a proportion of the BSi deposited at the seafloor (11% calculated from Fig. 1 in Tréguer et al., (2021) and calculated in Frings (2017) (doi.org/10.1007/s11631-017-0183-1)). Barents Sea stations therefore exhibit similar burial efficiencies to the global average, but lower than most values that are reported for continental shelf sediments.**